# COOL: EFFICIENT AND RELIABLE CHAIN-ORIENTED OBJECTIVE LOGIC WITH NEURAL NETWORKS FEEDBACK CONTROL FOR PROGRAM SYNTHESIS

## ABSTRACT

Program synthesis methods, whether formal or neural-based, lack fine-grained control and flexible modularity, which limits their adaptation to complex software development. These limitations stem from rigid Domain-Specific Language (DSL) frameworks and neural network incorrect predictions. To this end, we propose the **Chain of Logic (CoL)**, which organizes synthesis stages into a chain and provides precise heuristic control to guide the synthesis process. Furthermore, by integrating neural networks with libraries and introducing a **Neural Network Feedback Control (NNFC)** mechanism, our approach modularizes synthesis and mitigates the impact of neural network mispredictions. Experiments on relational and symbolic synthesis tasks show that CoL significantly enhances the efficiency and reliability of DSL program synthesis across multiple metrics. Specifically, CoL improves accuracy by 70% while reducing tree operations by 91% and time by 95%. Additionally, NNFC further boosts accuracy by 6%, with a 64% reduction in tree operations under challenging conditions such as insufficient training data, increased difficulty, and multidomain synthesis. These improvements confirm COOL as a highly efficient and reliable program synthesis framework.

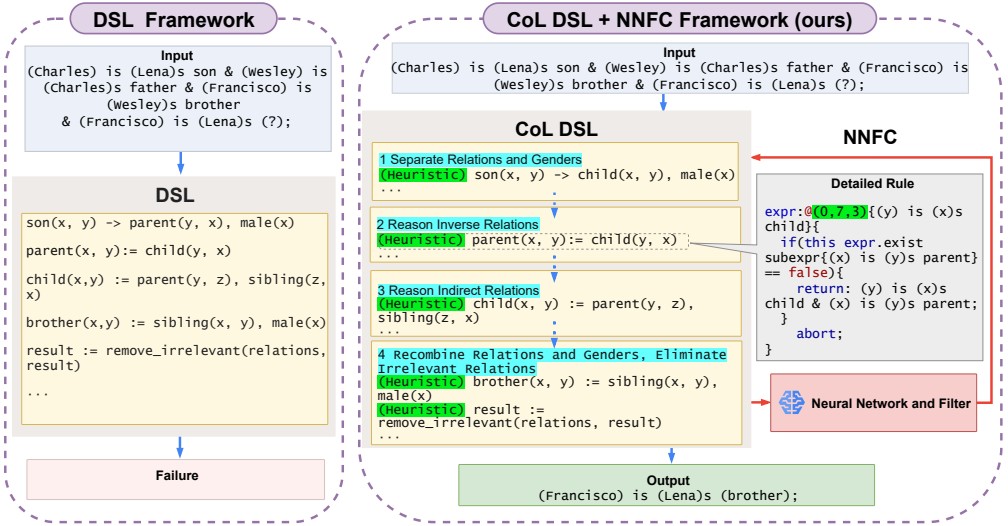

Figure 1: Chain-of-Logic (highlighted part) organizes the rule application into a structured sequence, enhancing the Domain-Specific Language (DSL) framework's ability to handle complex tasks. The Neural Network Feedback Control mechanism (red path) utilizes data during synthesis to improve the performance of the synthesis process dynamically.

# 1 INTRODUCTION

Program synthesis is becoming increasingly important in computer science for enhancing development efficiency Gulwani et al. (2017); Jin et al. (2024). Despite the effectiveness of current state-of-the-art methods in dealing with simple tasks, the complexity of modern software demands more advanced and sophisticated approaches Sobania et al. (2022).

To address these challenges, an effective solution must offer programmers fine-grained control and flexible modularity in the synthesis process Groner et al. (2014); Sullivan et al. (2001). First, fine-grained control tailors the synthesis path to specific tasks, ensuring the interpretability of the synthesis process. Secondly, flexible modularity enhances reusability and guarantees the quality of the entire program by ensuring the correctness of the modules Le et al. (2023).

However, these principles are often overlooked in current state-of-the-art program synthesis methods. For example, symbolic approaches such as SyGus Alur et al. (2013), Escher Albarghouthi et al. (2013), and FlashFill++ Cambronero et al. (2023) struggle to scale to complex tasks because their traversal-based Domain-Specific Language (DSL) framework lacks fine-grained control. A compensatory strategy involves using neural networks for

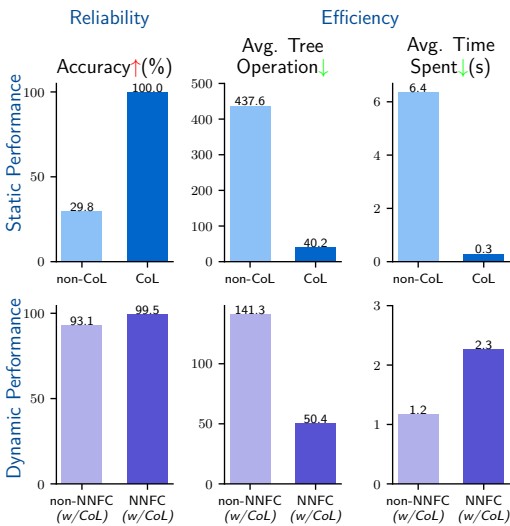

Figure 2: Performance Enhancements with CoL and NNFC. The CoL DSL surpasses non-CoL DSL in all metrics. While NNFC increases computation time due to neural network calls, it significantly boosts accuracy in dynamic experiments, enhancing reliability.

guidance or search space pruning, as seen in projects such as Neo Feng et al. (2018), LambdaBeam Shi et al. (2023a), Bustle Odena et al. (2020), DreamCoder Ellis et al. (2023), and Algo Zhang et al. (2023), but the control logic remains disconnected from the programmer. On the other hand, LLM-based projects like CodeGen Nijkamp et al. (2022), CodeX Finnie-Ansley et al. (2022), and Code Llama Roziere et al. (2023) allow programmers to control synthesis through prompt interactions. However, they lack modularity, as all tasks rely on the same LLM, making the logic vulnerable to biases in training data and leading to subtle errors that require manual verification. In summary, there is an urgent need for fine-grained control and flexible modularity to ensure the efficiency and reliability of these methods when tackling complex synthesis tasks.

In this paper, following the principles of fine-grained control and flexible modularity, we present **COOL (Chain-Oriented Objective Logic)**, a neural-symbolic framework for complex program synthesis. At the core of our approach, we introduce the **Chain-of-Logic (CoL)**, which integrates the functions of the activity diagram to enable fine-grained control Gomaa (2011). As illustrated in Figure 1, programmers can precisely organize rules into multiple stages and manage control flow using heuristics and keywords. Additionally, we leverage neural networks on top of CoL to dynamically fine-tune the synthesis process. For this purpose, we introduce **Neural Network Feedback Control (NNFC)** Turan & Jäschke (2024), which enhances future synthesis by learning from data generated during synthesis and suppresses neural network incorrect predictions through filtering. To ensure modularity, each neural network is bound with a specific CoL DSL, stored in separate library files for clear isolation and easy reuse. Thus, through the combination of CoL and NNFC, COOL achieves high efficiency and reliability when tackling complex synthesis tasks.

We conduct static experiments (constant domain and difficulty tasks, using pre-trained neural networks without further training) and dynamic experiments (mutative domain and difficulty tasks, where neural networks are created and continuously trained during the experiment) to evaluate the impact of CoL and NNFC on program synthesis. Figure 2 illustrates the significant improvements achieved by CoL and NNFC: In static experiments, CoL improves accuracy by 70%, while reducing

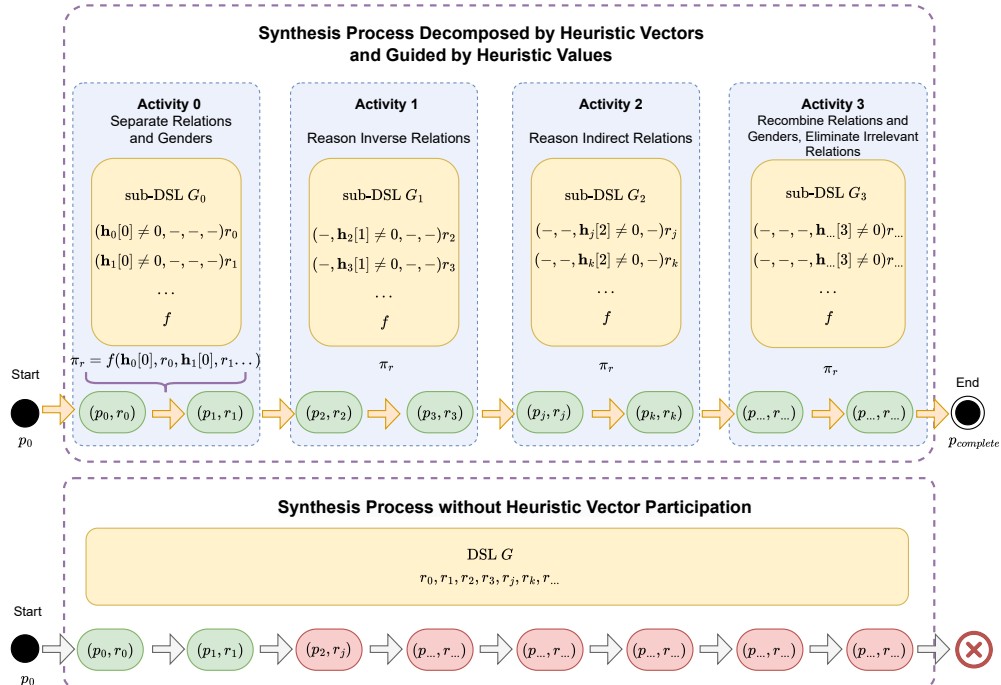

Figure 3: Heuristic Vectors and Heuristic Values in Chain-of-Logic. Heuristic vectors $\mathbf{h}$ decompose the DSL $G$ into multiple sub-DSLs $(G_0, G_1, G_2, G_3)$ based on whether the value of the component at the corresponding position is non-zero. These sub-DSLs correspond to the activities depicted in Figure 1 and operate on partial programs $p$ using rules $r$. The synthesis process for each activity is guided by rule application policies $\pi_r$, which are generated by a heuristic algorithm $f$ that uses heuristic values $\mathbf{h}[n]$ as input. In the experiments conducted in this paper, we adopt the A* algorithm and treat the heuristic value $\mathbf{h}_i[n]$ as a reward for applying the rule $r_i$ during activity $n$. Consequently, a higher heuristic value positively influences the rule's application.

tree operations by 91% and time by 95%. In dynamic experiments, NNFC further increases the accuracy by 6%, with a 64% reduction in tree operations. The results underscore that achieving fine-grained control and flexible modularity can greatly improve efficiency and reliability in DSL program synthesis.

The contributions of our work are as follows:

1. We propose the **Chain-of-Logic (CoL)**, which enables fine-grained control in complex program synthesis by structuring rule applications into distinct and manageable stages.

2. We further introduce **Neural Network Feedback Control (NNFC)**, a dynamic correction mechanism for CoL that continuously learns from the synthesis process, ensuring modularity by pairing neural networks with specific CoL DSLs.

3. We present **COOL**, an efficient and reliable neural-symbolic framework for complex program synthesis, combining the strengths of CoL and NNFC to achieve fine-grained control and flexible modularity in DSL-based synthesis.

## 2 METHOD

In this section, we detail the implementation of CoL and NNFC, outlining the principles that ensure high efficiency and reliability for complex program synthesis tasks.

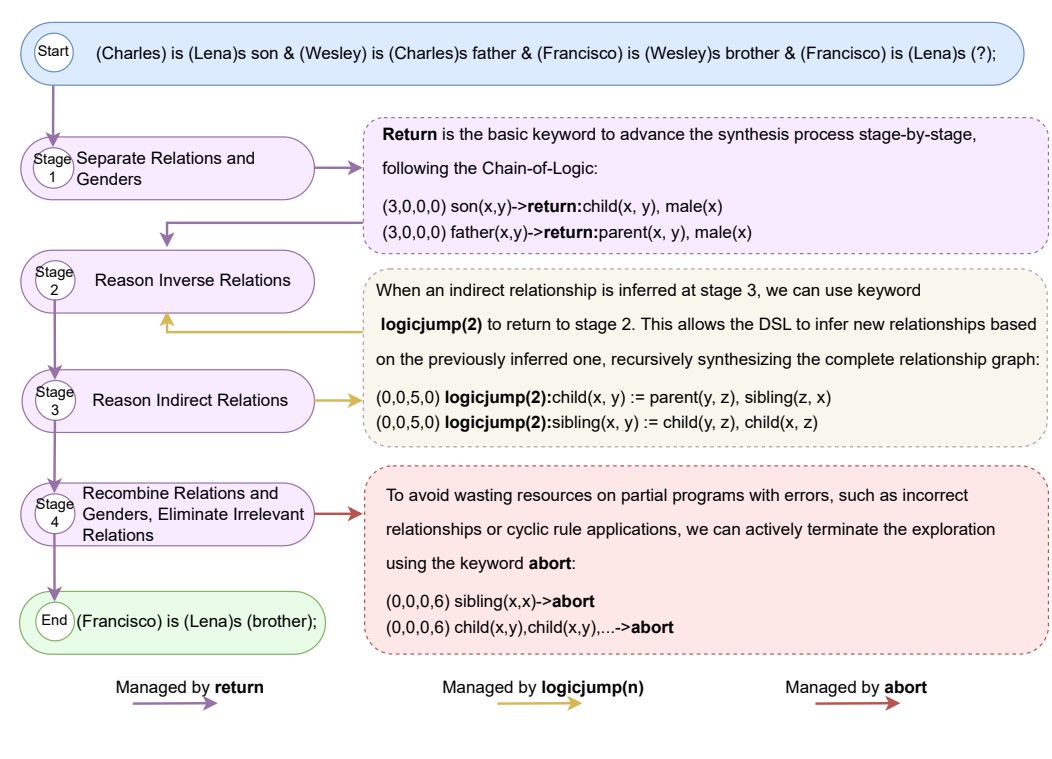

Figure 4: Keywords in Chain-of-Logic. In this illustrative CoL DSL, each node represents a stage or activity where a set of rules can be applied to generate partial programs. The flow between stages is managed by keywords **return**, **logicjump(n)**, and **abort**, allowing for the implementation of complex control flow in program synthesis.

## 2.1 CHAIN-OF-LOGIC (CoL)

Activity diagrams, widely used in software engineering, effectively describe how an initial state transitions to a final state through multiple activities. This feature aligns with the DSL-based program synthesis process. A DSL, defined as a context-free grammar, converts partial programs with nonterminal symbols into complete programs by applying given rules. However, as the rule set grows, DSL becomes inefficient in exploring partial programs. To enhance the efficiency of DSL, the Chain-of-Logic, drawing inspiration from activity diagrams, organizes the synthesis process into a sequence of manageable activities.

CoL improves the synthesis workflow of the DSL with two key features: *heuristic vectors* and *keywords*. As shown in Figure 3, heuristic vectors decompose DSL into multiple sub-DSLs that are consistent with the activity flow by dividing the rule applications scope. Within each activity, the DSL solver uses heuristic values to perform efficient program synthesis. For example, in Figure 1, a rule with the heuristic vector (0,7,3) belongs to sub-DSLs in activities 2 and 3 with heuristic values of 7 and 3, respectively. By dynamically pruning the search space and providing search guidance, heuristic vectors promote program synthesis efficiency.

Second, as illustrated in Figure 4, CoL introduces three keywords—return, logicjump(n), and abort—to dynamically manage state transition within or between activities during synthesis:

1. **return**: Ends the current rule, staying within current activity or advancing to following activities.

2. **logicjump(n)**: Jumps directly to the activity $n$, enabling branching and loops within activity flow.

3. **abort**: Terminates the current synthesis branch, pruning the search space.

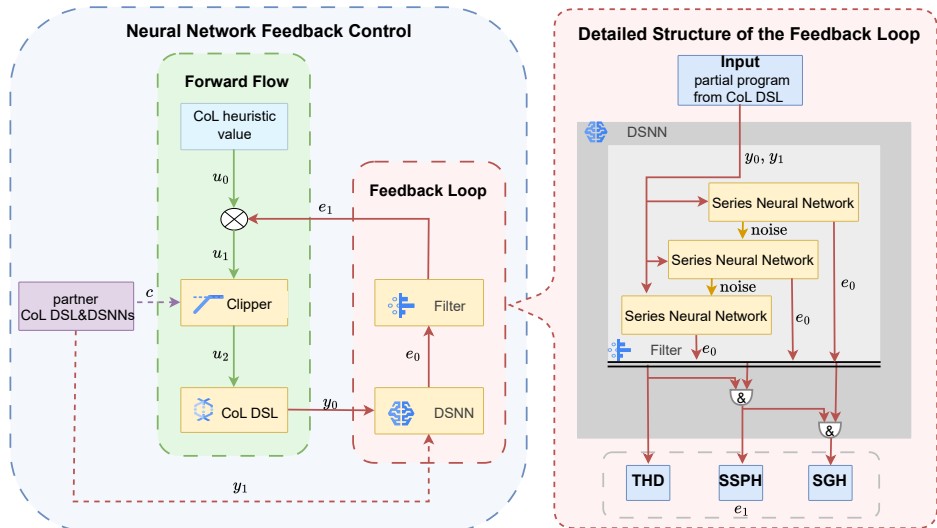

Figure 5: Neural Network Feedback Control. The left side illustrates the complete control loop of NNFC. In the forward flow (green path), heuristic values $u$ guide the synthesis process as control signals. In the feedback loop (red path), the **DSNN (Domain-Specific Neural Network, the neural network paired with a DSL)** generates initial error signals $e_0$ from partial programs $y$. These singals are then filtered to produce high-quality error signals $e_1$, which adjust the initial heuristic values $u_0$. In multidomain synthesis, the CoL DSL and DSNN from the self-domain use partner domain information (dashed path) to clarify tasks and avoid competition, ensuring modularity. The right side details the feedback loop: The DSNN comprises multiple neural networks coupled in series via noise signals, with each network generating its own error signal $e_0$, then these signals with large discrepancies are filtered, retaining the final high-quality error signals $e_1$.

Based on the principle of activity diagrams, CoL provides fine-grained control through heuristic vectors and keywords. This systematic approach enhances the efficiency of DSL synthesis.

## 2.2 NEURAL NETWORK FEEDBACK CONTROL (NNFC)

While CoL enables programmers to fine-tune the synthesis process, the control flow may lack detail or vary by task. To this end, **Neural Network Feedback Control (NNFC)** dynamically refines control flow through feedback from neural networks, improving precision and adaptability. However, neural networks present the risk of generating incorrect predictions, threatening reliability.

Therefore, a robust control flow in NNFC is crucial to ensuring overall performance. As illustrated in Figure 5, NNFC enhances the CoL DSL in the following ways: In the forward flow, the Clipper prioritizes control signals aligned with DSNN guidance by capping any inconsistent signals, while the CoL DSL applies rules based on the adjusted heuristic values. Meanwhile, in the feedback loop, the DSNN generates error signals from partial programs across domains. To suppress the impact of mispredictions, the Filter refines these signals before they influence the forward flow.

The quality of the signals generated in the feedback loop directly determines the effectiveness of NNFC. If the error signals are of poor quality, NNFC may not only fail to provide additional improvements but also degrade CoL DSL performance. We ensure the error signal quality through an inner coupling structure within DSNN. As shown in Figure 5 (right), during synthesis tasks, DSNN processes partial programs using a series of sequentially connected neural networks. Each neural network takes both the partial programs and intermediate results from the preceding neural network as input, generating its own predictions. When errors occur in earlier networks, they propagate downstream as noise signals, amplifying at each stage. The difference in the outputs between these neural networks is positively correlated with the accumulated error. To mitigate this, we set a threshold to filter out signals with a significant difference in outputs. Finally, DSNN uses passed signals to generate multi-head outputs to fine-tune the forward flow:

1. **Task Detection Head (TDH)**: Improves modularity by determining whether the partial program contains components that the CoL DSL can process.

2. **Search Space Prune Head (SSPH)**: (Active when TDH is true) Evaluate the feasibility of synthesizing the final complete program from the current partial program, and CoL DSL will avoid exploring infeasible spaces.

3. **Search Guidance Head (SGH)**: (Active when both TDH and SSPH are true) Guides the CoL DSL in applying the most promising rules to the partial program.

By adopting filtering and multi-head outputs, the feedback loop delivers high-quality error signals to the forward path, ensuring that NNFC enhances the synthesis process on top of CoL.

## 3 EXPERIMENTS

We conduct the experiments in two stages to evaluate the improvements introduced by CoL to DSL and to assess how NNFC further enhances performance. First, we carry out static experiments under fixed conditions, including task domain, difficulty level, and neural network. These controlled conditions allow us to accurately measure CoL's impact on performance. Next, we proceed with dynamic experiments, where conditions vary throughout. This dynamic setup evaluates NNFC's ability to improve reliability under changing situations.

### 3.1 EXPERIMENTAL SETUP

Improvements of DSL by CoL and NNFC is evaluated across benchmarks using various metrics.

**Benchmarks.** We evaluate CoL and NNFC using relational and symbolic tasks with varying difficulty levels, as detailed in Table 1. Specifically, the relational tasks are drawn from the CLUTRR Sinha et al. (2019) dataset, where the goal is to synthesize programs that capture specific target relationships based on human common-sense reasoning. In contrast, the symbolic tasks are generated by GPT Achiam et al. (2023). They involve synthesizing standard quadratic equation programs from non-standard quadratic forms by performing manual calculation steps. Although these tasks are simple for humans, they serve as a straightforward demonstration of how fine-grained control, derived from programmer expertise, can significantly improve program synthesis efficiency.

**Metrics.** Besides accuracy, we also focus on the following points: **(1) CPU Overhead** is assessed by the number of tree operations required for synthesis. **(2) Memory overhead** is assessed by the number of transformation pairs (a partial program paired with the rule to be applied)[1]. **(3) GPU Overhead** is measured by the number of neural network invocations. **(4) Time overhead** is referenced by the actual time spent on program synthesis tasks. **(5) Filtering Performance** is evaluated by the attenuation ratio of invalid to passed neural network predictions.

**Chain-of-Logic.** We utilize the CoL approach to enhance DSL by making the synthesis process more in line with human problem-solving strategies. For relational tasks, by mirroring the way humans typically reason about family relationships, CoL organizes the synthesis process into stages illustrated in Figure 4. For symbolic tasks, CoL structures the DSL to follow the manual quadratic equation simplification strategy, with stages such as expanding terms, extracting coefficients, permuting terms, and converting equations to standard form. The specific CoL DSL configurations are shown in Table 2, where the significant differences in DSLs highlights the generality of CoL.

---

[1]Each partial program must be completed with at most 1000 transformation pairs, though this may exceed 1000 if additional tasks are generated during synthesis.

Table 1: Benchmark configurations. Relational benchmarks are divided into easy and difficult groups based on the number of relationship edges, while symbolic benchmarks are based on the number of nodes in the tree.

| Benchmark Type | Difficulty Level A | Difficulty Level B |
|:---:|:---:|:---:|
| relational | 300 tasks with 3 edges | 200 tasks with 4 edges |
| symbolic | 300 tasks with around 5 nodes | 200 tasks with around 9 nodes |

Table 2: CoL DSL configurations. The DSL for relational benchmarks has a limited search space and shorter CoL, facing challenges from numerous production rules leading to larger trees. Conversely, the DSL for symbolic benchmarks offers an unlimited search space with a longer CoL, but the many permutation rules increase the risk of cyclic rule applications.

| Benchmark | Rules | | | | | Length of CoL |
|---|---|---|---|---|---|---|
| | Total | Production Rules | Reduction Rules | Recursive Rules | Permutation Rules | |
| relational | 40 | 36 | 2 | 16 | 0 | 4 |
| symbolic | 55 | 17 | 26 | 3 | 11 | 7 |

**Groups.** We use multiple groups to comprehensively evaluate CoL and NNFC (as shown in Table 3). First, in static experiments, we evaluate CoL by comparing DSL groups with and without CoL enhancements. Second, to isolate the impact of heuristic vectors—both as guides and as structuring tools for rule application—we create groups enhanced only by heuristic values. Third, we introduce groups enhanced by neural networks to assess whether combining CoL with neural networks yields better results and to explore the filtering effect of the inner coupling structure. In dynamic experi-

Table 3: Group configurations. Groups marked with ★ are the main experiments, those with ☆ are for ablation and extended experiments, and the unmarked group is the baseline.

| Group | Experiment | Pretrained DSNN | NNFC | Inner Coupling Structure |
|---|---|---|---|---|
| DSL | static | | | |
| ☆DSL (Heuristic) | static | | | |
| ★CoL DSL | static, dynamic | | | |
| ☆DSL+NN | static | ✓ | | |
| ☆DSL (Heuristic)+NN | static | ✓ | | |
| ☆ CoL DSL+NN | static | ✓ | | |
| ☆CoL DSL+NNFC | dynamic | | ✓ | |
| ☆DSL+NN (Cp) | static | ✓ | | ✓ |
| ☆DSL(Heuristic)+NN (Cp) | static | ✓ | | ✓ |
| ☆CoL DSL+NN (Cp) | static | ✓ | | ✓ |
| ☆CoL DSL+NN (Cp) | static | ✓ | | ✓ |
| ★CoL DSL+NNFC (Cp) | dynamic | | ✓ | ✓ |

ments, we design control groups with and without NNFC to evaluate its impact. Additionally, we include a group without the inner coupling structure to confirm its necessity.

**Environment.** Experiments are carried out on a computer equipped with an Intel i7-14700 processor, a GTX 4070 GPU, and 48GB RAM.

## 3.2 STATIC EXPERIMENTS

We start with static experiments. With the task domain, difficulty level, and neural network conditions unchanged in each group, a series of controlled experiments confirm that CoL has remarkably boosted DSL program synthesis in all metrics.

The results in Table 4 clearly demonstrate that **CoL significantly improves accuracy while minimizing overhead**. Most notably, CoL improves the accuracy of the DSL from less than 50% to 100% across both relational and symbolic benchmarks. Additionally, CoL achieves remarkable reductions in relational tasks, cutting tree operations by 90%, transformation pairs by 88%, and time by 95%. Similarly, in symbolic tasks, CoL reduces tree operations by 92%, transformation pairs by 96%, and time by 97%. These findings showcase CoL's substantial impact on improving performance across all key metrics.

Table 4: Static performance of DSL and CoL DSL for relational and symbolic tasks. CoL DSL significantly outperforms DSL in all metrics.

| Benchmark | Group | Accuracy↑ (%) | Avg. Tree Operation↓ | Avg. Transformation Pair↓ | Avg. Time Spent↓(s) |
|---|---|---|---|---|---|
| relational | DSL | 11.3 | 463.9 | 1432.2 | 9.43 |
| | CoL DSL | **100.0** | **46.6** | **177.8** | **0.48** |
| symbolic | DSL | 48.3 | 411.2 | 2285.3 | 3.31 |
| | CoL DSL | **100.0** | **33.8** | **92.7** | **0.11** |

Further ablation and extension experiments clarify the sources of CoL's enhancement, confirm CoL's effective integration with neural networks, and explore when filtering via inner coupling structures is most beneficial. Our findings are as follows:

First, **CoL's enhancement stems from both heuristics and structured rule application stages**. As illustrated in Figure 6, the DSL (Heuristic) group outperforms the DSL group in most metrics, and the CoL DSL group significantly surpasses DSL (Heuristic) in all metrics. Such results indicate that CoL positively impacts synthesis by guiding and structuring rule application. Moreover, on top of guidance, the structured rule application stages achieve greater improvement.

Second, **integrating CoL with neural networks further improves the search efficiency.** As shown in Figure 6, despite additional GPU and time overhead, the top-performing CoL DSL + NN group reduces tree operations by 43% and transformation pairs by 19% in relational tasks compared to the CoL DSL group. In symbolic tasks, the CoL DSL + NN (Cp) group reduces tree operations by 64% and transformation pairs by 46%. The results showcase that neural networks can further narrow the search space for program synthesis beyond CoL. Importantly, the group with the inner coupling structure outperforms non-neural groups in both tasks. In contrast, the group without it presents an accuracy decline in symbolic tasks, validating the structure's role in improving reliability.

Third, **the inner coupling structure is more effective when error tolerance is low**. As indicated in Figure 6, for symbolic tasks, CoL DSL-based groups with the inner coupling structure significantly outperform those without it. However, for relational tasks and DSL-based groups (without CoL or heuristic), those without such structure perform better. This difference indicates that the filtering effect of the inner coupling structure comes at a cost: it filters out both incorrect and correct predictions. So, its effectiveness depends on the positive impact of eliminating incorrect predictions outweighing the loss of correct ones. Therefore, for relational tasks with a limited search space and DSL-based groups with higher error tolerance, the cost of filtering outweighs the benefit. However, in symbolic tasks, where avoiding errors is more critical, CoL DSL-based groups benefit significantly from the inner coupling structure.

## 3.3 DYNAMIC EXPERIMENTS

Static experiments confirm CoL's improvements on DSL and its enhancement with neural networks. However, real-world program synthesis involves varying task domains and difficulty, facing the risk of neural network mispredictions due to underperformance. Therefore, we introduce these factors in dynamic experiments to evaluate how NNFC further improves the performance of CoL DSL.

The results in Table 5 confirm that **NNFC significantly enhances the reliability of CoL DSL in challenging conditions.** As task difficulty increases and multidomain scenarios emerge, the accuracy of the CoL DSL group declines compared to its performance in static experiments. However, the NNFC-enhanced group maintains an accuracy of at least 99%, demonstrating its strong reliability in challenging situations. Additionally, compared with the original CoL DSL group, it reduces tree operations by 22% and transformation pairs by 14%. For symbolic tasks, despite the added time for neural network invocations, the NNFC-enhanced group still shortens the time spent by 21%.

Further ablation experiments confirm that **reliability provided by NNFC primarily stems from the filtering effect of the inner coupling structure.** As shown in Figures 7 and 9, the inner cou-

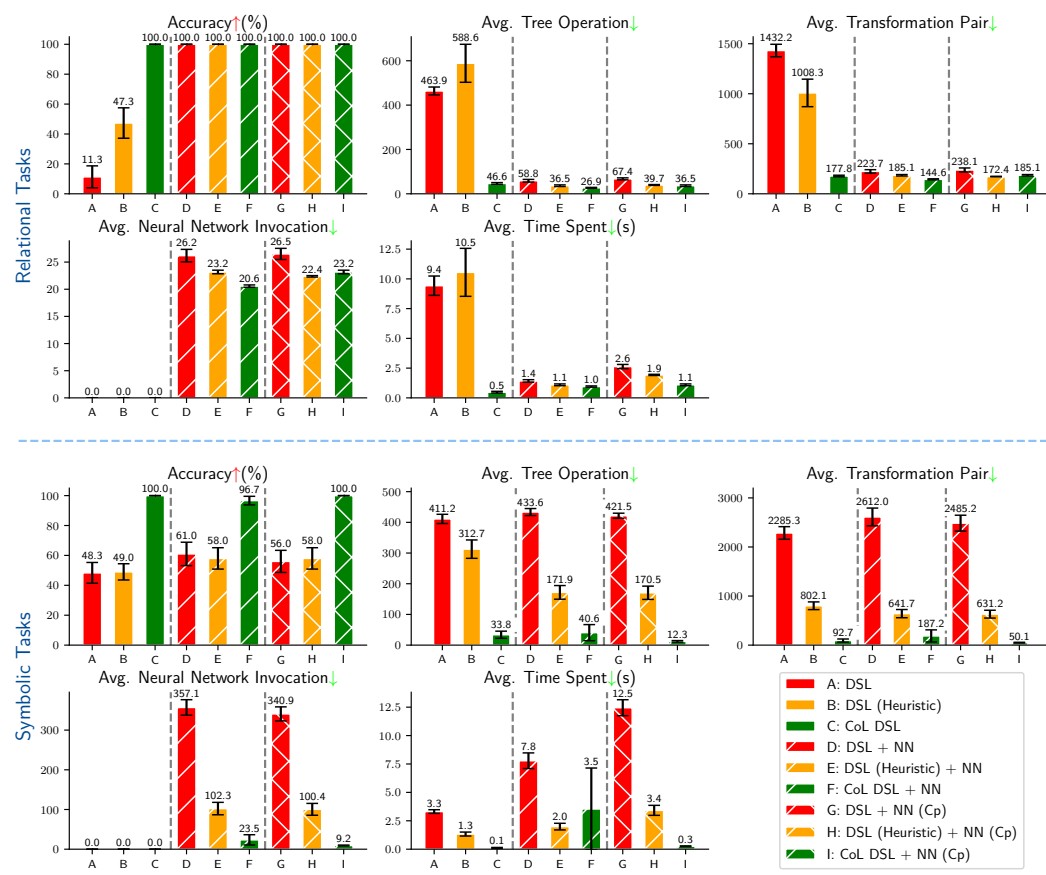

Figure 6: Static performance on relational and symbolic tasks at difficulty level A. CoL DSL-based groups outperform DSL (Heuristic) and DSL groups. Performance varies for DSNN-enhanced groups with the inner coupling structure. Error bars show 95% confidence intervals across 6 batches.

Table 5: Dynamic performance of CoL DSL and CoL DSL+NNFC(Cp). NNFC significantly improves the dynamic performance of CoL DSL in accuracy, tree operations, and transformation pairs.

| Bench-mark | Group | Accuracy (%) | Avg. Tree Operation | Avg. Trans-formation Pair | Avg. Neural Network Invocation | Avg. Time Spent (s) |
|---|---|---|---|---|---|---|
| relational | CoL DSL | **100.0** | 70.0 | 259.8 | **0** | **1.05** |
| | CoL DSL+NNFC (Cp) | **100.0** | **54.6** | **224.5** | 21.7 | 2.08 |
| symbolic | CoL DSL | 82.6 | 233.5 | 977.1 | **0** | 1.42 |
| | CoL DSL+NNFC (Cp) | **99.4** | **50.3** | **222.2** | 21.6 | **1.12** |
| multi-domain | CoL DSL | 97.5 | 115.2 | 367.6 | **0** | **0.99** |
| | CoL DSL+NNFC (Cp) | **99.0** | **45.6** | **250.5** | 72.84 | 3.91 |

pling structure reduces the occurrence of accuracy declines due to DSNN mispredictions by 94%. Additionally, the dynamic performance reveals how the inner coupling structure enhances NNFC:

In the scenarios where a DSNN underperforms due to issues such as insufficient training data Mikołajczyk & Grochowski (2018) (as seen in Figure 7, tasks 51-100), inadequate generalization to more challenging tasks Yosinski et al. (2014); Wei et al. (2019) (Figure 7, tasks 301-350), and catastrophic forgetting when tasks from a new domain are learned Kirkpatrick et al. (2017);

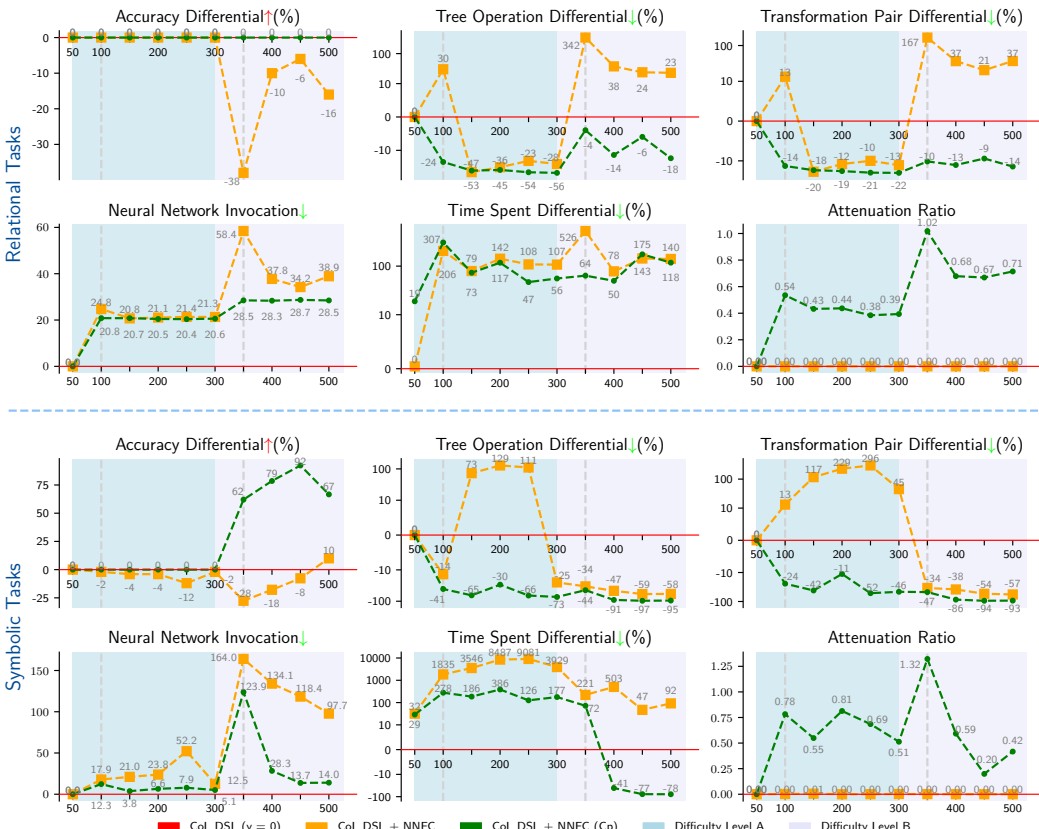

Figure 7: Dynamic performance differential to CoL DSL in singledomain tasks. The NNFC group without the inner coupling structure shows 12 accuracy declines across 20 batches, while the group with the structure shows none. Each batch consists of 50 tasks, and NNFC continuously trains DSNNs using generated data after each batch, starting from scratch.

Van de Ven & Tolias (2019) (Figure 9, tasks 1-100), incorrect predictions lead the actual synthesis path to deviate from the CoL, which in turn causes inefficiency and reduced accuracy. During these phases, for NNFC with the inner coupling structure, the attenuation ratio spikes, indicating that a large percentage of neural network predictions are filtered out. Consequently, the inner coupling structure ensures that the synthesis process adheres to the CoL, effectively mitigating the negative impact of DSNN mispredictions and enhancing reliability.

As the DSNN improves and reaches a relatively stable state (as seen in Figure 7, tasks 101-300, 351-500, and Figure 9, tasks 101-400), the attenuation ratio shows a decreasing trend accordingly. This adaptive adjustment demonstrates how the inner coupling structure dynamically regulates the DSNN's impact, leveraging neural network contributions while mitigating risks to ensure both efficiency and reliability in program synthesis.

## 4 CONCLUSION

We explored fine-grained control and flexible modularity for complex program synthesis through the Chain-Oriented Objective Logic (COOL) framework. Inspired by activity charts and control theory, we developed Chain-of-Logic (CoL) and Neural Network Feedback Control (NNFC) to achieve these goals. Static and dynamic experiments across relational, symbolic, and multidomain tasks demonstrated that COOL offers strong efficiency and reliability. We believe that continued research and refinement of CoL and NNFC will inspire advancements not only in program synthesis but also in broader areas of neural network reasoning.

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

# A  DSL PROGRAM SYNTHESIS IN COOL

COOL adopts a top-down synthesis strategy that converts input partial programs with nonterminals into complete programs by applying a sequence of well-defined transformation rules.

## A.1  INPUT PROGRAM

In the relational reasoning tasks, the input is COOL code such as Code A.1:

### Code A.1: Relational Reasoning Task Input Program

```
(Wesley) is (James)s son & (Martha) is (Wesley)s daughter &
(Hugh) is (Martha)s uncle & (Hugh) is (James)s ($relation);
```

where $ specifies the nonterminal, indicating that the DSL solver needs to synthesize a complete program that calculates the correct value for `relation` (the relationship between `Hugh` and `James`) in order to satisfy the given specification.

The symbolic task input is The input for symbolic tasks is as follows (Code A.2):

### Code A.2: Symbolic Reasoning Task Input Program

```
$x^2 + 4*$x == 3;
```

Similarly, the DSL solver needs to generate a complete program that calculates the value of `x`.

## A.2  OUTPUT PROGRAM

As shown in Code A.3, for family relationship reasoning tasks, the synthesized program is:

### Code A.3: Relational Reasoning Task Output Program

```
relation = "son";
```

For symbolic reasoning tasks, the generated output program is shown in Code :

### Code A.4: Symbolic Reasoning Task Output Program

```
Invoke Quadratic Solution Formula( a=1, b=4, c=-3, x ); // a,
b, c are coefficients in "1*x^2 + 4*x + (-3) == 0"
```

In reality, the program synthesis takes place at the intermediate representation level (see Appendix Q), and Codes A.3 A.4 are provided for explanatory purposes.

## A.3  TRADITIONAL DSL

A DSL, defined as a context-free grammar:

$$G = \{V, \Sigma, R, S\}, \tag{1}$$

where $V$ is the set of non-terminal symbols, $\Sigma$ is the set of terminal symbols, $R$ is the set of rules, and $S$ is the starting symbol (in this context, it is a partial program). The DSL's derivation process converts partial programs with nonterminal symbols into complete programs by applying given rules.

The synthesis process for traditional DSLs involves iteratively transforming partial programs into complete programs by applying a series of rules. Each partial program ($p$) and the corresponding rule ($r$) to be applied to it form a transformation pair $(p, r)$. When a rule is applied, it modifies

the syntax tree of the partial program through what we refer to as a tree operation. The synthesis process consists of a series of transformation pairs connected by tree operations, which is known as a synthesis path or trajectory. These paths are classified into three kinds:

- **Feasible Path**: Leads to a complete program.
- **Infeasible Path**: Proven to be unable to synthesize a complete program.
- **Unfinished Path**: Still in progress.

To clarify key concepts involved in the synthesis process, we provide the following definitions of terms:

- **Tree Operation/Manipulation**: Refers to the modification of the syntax tree of a partial program during the synthesis process. It is essential for transforming partial programs into complete programs and has an associated CPU cost.
- **Transformation Pair** $(p, r)$: A combination of a partial program and a rule to be applied. It records the explored space and possible exploration directions, requiring memory storage.
- **Synthesis Path/Trajectory**: A sequence of transformation pairs, $\{(p_0, r_0), (p_1, r_1), \dots\}$, representing the process of transforming a partial program into a complete one. Its function is to track the entire synthesis process, whether it leads to a feasible, infeasible, or unfinished path.

### A.4    CoL DSL

Compared with traditional DSLs that apply rules to input programs without a clear destination to synthesize output programs, the Chain-of-Logic (CoL) allows the programmer to outline the flow of activities to synthesize the complete program from an initial partial program. For example, in Figure 3, the activity flow is:

Start $\rightarrow$ 1 Separate Relations and Genders $\rightarrow$ 2 Reason Inverse Relations $\rightarrow$ 3 Reason Indirect Relations $\rightarrow$ 4 Recombine Relations and Genders, Eliminate Irrelevant Relations $\rightarrow$ End.

Each activity in the synthesis process has a corresponding sub-DSL decomposed from the original DSL for transforming the program from one state to another.

Therefore, a CoL DSL with $n$ activities can be defined as multiple sub-DSLs in series:

$$\text{CoL } G = \{G_1, G_2, \dots, G_n\} \tag{2}$$

A sub-DSL for activity $i$ is defined as:

$$G_i = \{V, \Sigma, \{(r, \mathbf{h}, k) \mid \mathbf{h}[n] \neq 0 \text{ and } r \in R\}, S, f\} \tag{3}$$

where $\mathbf{h} = (h_1, h_2, \dots)$ in $(r, \mathbf{h}, k)$ represents the heuristic vector bound with $r$, and the components are termed as heuristic values. $h[n]$ represents the $n$-th component of $h$, which is the effective heuristic value of rule $r$ in activity $n$. $\mathbf{h}[n]$ is a parameter of the sub-DSL's program synthesis algorithm $f$. It guides the direction of program synthesis by affecting the application decisions of the rules bound to it, thereby improving synthesis efficiency and accuracy. The specific role of the heuristic value is determined by the search algorithm $f$ used by the DSL program synthesis. To conduct controlled variable experiments, we regard the heuristic value as a reward (negative cost) and use the A* algorithm as the search algorithm for all DSLs, which means that in activity $n$, under the same circumstances, the rule with a larger $\mathbf{h}[n]$ will be applied first, and the derived program will also be considered promising and will be explored further with priority.

$k$ represents keyword(s) in a rule's specific logic, which controls the program state transition within an activity or between activities.

The Chain-of-Logic provides a comprehensive methodology for achieving fine-grained control over DSL program synthesis. This approach allows programmers to explicitly break down the synthesis process into distinct phases or activities, with each activity corresponding to a specific sub-DSL.

### A.5 CoL DSL Synthesis Process

The synthesis process in CoL DSLs is conducted through multiple stages, each corresponding to a defined activity. These stages operate sequentially, gradually refining the program through well-defined transformations. The key difference from traditional DSL synthesis is that, except for the sub-DSL of the final activity, intermediate sub-DSLs are allowed to generate partial programs, which are passed on to subsequent activities for further processing.

Each stage in the CoL DSL synthesis process focuses on a specific aspect of synthesis to transform the program incrementally. For instance, in Figure 4:

- In the first activity, relations and genders are separated, breaking down the initial partial program into simpler components for easier processing.
- The second activity reasons about inverse relationships, further structuring the intermediate program by identifying and processing inverse connections.
- The third activity deals with indirect relationships, providing additional context to relationships identified in earlier stages.
- The final activity recombines relations and genders while eliminating irrelevant relations to produce a fully synthesized and optimized program.

During each activity, the synthesis process leverages heuristic values to prioritize rule application, focusing on areas that are more likely to lead to successful outcomes. Additionally, in intermediate activities, since we cannot judge whether the synthesis process is correct based on whether complete programs are generated, guidance based on heuristic values and synthesis flow control using keywords are pivotal.

## B  NEURAL NETWORKS IN COOL

COOL has an integrated machine learning system that automatically collects generated data and conducts training and prediction tasks for neural networks in the Domain-Specific Neural Network (DSNN).

### B.1 Data Collection and Combination for Training

The neural networks leverage the transformation pairs $(p, r)$ in the synthesis paths to train various heads.

To train the neural networks in DSNN bound with a DSL for program synthesis tasks of type $T$, COOL builds the dataset as follows:

**Task Detection Head (TDH)**: This head distinguishes whether the input partial program belongs to type $T$. This is a binary classification task. The partial programs from type $T$ program synthesis paths are collected as positive examples (proportion: 67%), while partial programs from other synthesis paths and built-in function calls are collected as negative examples (proportion: 33%).

**Search Space Prune Head (SSPH)**: After determining that the program is of type $T$, this head identifies whether the input partial program is feasible to synthesize into a complete program. This is also a binary classification task. Programs from feasible synthesis paths are collected as positive examples (proportion: 67%), while programs from infeasible synthesis paths are collected as negative examples (proportion: 33%).

**Search Guidance Head (SGH)**: After determining that the input is a feasible type $T$ partial program, this head generates rule features to guide the DSL solver in applying rules to the partial program. This includes a series of classification and regression tasks.

### B.2 Neural Network Input

As shown in Figure 5, there are three neural networks in a DSNN. Each network (labeled A, B, and C in their sequential order) takes a partial program as input. The input partial program is

Table 6: Input features of neural networks in DSNN. Each entry specifies the feature, its size, and the neural networks it pertains to, along with a description of its role. These features contribute to the neural network's understanding of the syntax tree's structure and semantics, aiding in the accurate synthesis of programs.

| Feature | Feature Size | Neural Network | Signification |
|---|---|---|---|
| grounded | 2 | A, B, C | The node is in a fully specified expression. |
| domain | 1 | A, B, C | Domain of the subtask represented by the subtree where the node is located. |
| root | 2 | A, B, C | The tree representing the subtask is rooted at this node. |
| non-terminal | 2 | A, B, C | The node is a non-terminal. |
| type | 1 | A, B, C | Type of the node. |
| identifier | 1 | A, B, C | Identifier of the node. |
| string | 1 | A, B, C | The node contains a string as the immediate value. |
| number | 1 | A, B, C | The node contains a number as the immediate value. |
| operator | 1 | A, B, C | The node is an operator. |
| current stage | 1 | A, B, C | Current CoL stage (valid when this node is grounded). |
| operand position | 3 | A, B, C | Placement of nodes in a binary operation tree (left operand node, right operand node, operation node). |
| applied (SGH) | 1 | B, C | A rule is applied to the subtree rooted at this node (derived from the output feature "**jumps**" of the previous neural network). |
| next stage (SGH) | 1 | C | The CoL stage to advance to after applying the rule (derived from the output feature "**next stage**" of the previous neural network). |

represented at the intermediate representation (IR) level in the form of Three-Address Code (TAC) (see Appendix Q), allowing program synthesis to be conducted without the constraints of specific DSL syntax or the machine code format of the execution platform Sujeeth et al. (2014). The TAC is then transformed into a graph representation for input to neural networks.

In the serial coupling structure of DSNN, network B is the downstream neural network of A and uses the output of the SGH head from A as part of its input. Similarly, network C is the downstream neural network of B and uses the output of the SGH head from B as part of its input. This serial coupling enables each downstream network to accumulate the error produced by the upstream network, making incorrect predictions more obvious.

The specific input features are shown in Table 6.

### B.3 NEURAL NETWORK OUTPUT

The corresponding relationships between the output features and TDH, SSPH, GSH are shown in Table 7, and all neural networks produce similar outputs to allow for comparison.

### B.4 NEURAL NETWORK STRUCTURE

As TAC embodies both the graphical properties of a syntax tree and the sequential properties of execution, the design of the neural network must be capable of capturing these dual characteristics.

The detailed layer architecture of neural networks in DSNN is illustrated in Figure 8. The processing flow consists of the following steps:

1. **Embedding Node Features:** We start by employing embedding layers with learning capabilities. These layers convert categorical inputs into dense, continuous vectors, which enhances the stability and efficiency of subsequent processing layers Hrinchuk et al. (2019).

Table 7: Output features of neural networks in DSNN. These features provide comprehensive optimizations for CoL DSL during program synthesis, including task detection, search space pruning, and search guidance.

| Feature | Feature Size | Neural Network | Signification |
|---------|-------------|----------------|---------------|
| domain (TDH) | 2 | A, B, C | Relevance of task domains to DSNN. |
| feasibility (SSPH) | 2 | A, B, C | Feasibility of synthesizing the complete program. |
| jumps (SGH) | max_tree_depth*3 | A, B, C | The path from the tree's root to the subtree's root where the rule is applied (jump left, right, or stop in each step). |
| next stage (SGH) | 1 | A, B, C | The CoL stage to advance to after applying the rule. |
| heuristic sign (SGH) | 2 | A, B, C | Sign of the rule's heuristic value. |
| heuristic value (SGH) | 1 | A, B, C | Rule's heuristic value. |
| expression (SGH) | 2 | A, B, C | Type of rule's head (expression or terminal). |

2. **Graph Feature Extraction:** Next, we use a Graph Neural Network (GNN) to extract graph features from each line of TAC code Drori et al. (2022); Wu et al. (2022). To adaptively extract intricate details such as node types, graph attention (GAT) layers are applied after the embedding layers Velickovic et al. (2017).

3. **Sequential Feature Processing:** We adopt Long Short-Term Memory (LSTM) networks to capture the sequential features inherent in TAC Chen et al. (2021); Nye et al. (2020). Recognizing the equal importance of each TAC line, bidirectional LSTM layers are employed following the GAT layers to enrich the contextual understanding Huang et al. (2015).

4. **Multi-Head Output:** Finally, the processed data is channeled through multiple output layers to prevent task interference and ensure clarity in results.

Figure 5 (right) illustrates using three neural network units arranged in series to construct the internal coupling structure of DSNN. Labeling these neural networks with A, B, and C in order of their sequence, Table 6 details the specific input features for each network: Neural network B receives its input feature "applied" from network A's output feature "jumps," while network C's input features "applied" and "next stage" are derived from the output features "jumps" and "next stage" of network B. The output features of three neural network units are consistent and comparable, Table 7 presents the output features of the neural networks.

B.5 PREDICTION FILTERING

By comparing the output differences of the heads, we can determine whether there are possible prediction errors and filter out the prediction results. For classification tasks, we directly compare whether the outputs are the same. For regression tasks, we set a tolerance threshold (10%) for the difference.

B.6 ACTING ON THE SYNTHESIS PROCESS

B.6.1 SINGLE DSL PROGRAM SYNTHESIS

As shown in Figure 3, the heuristic value of a rule affects its application, and the prediction results of the neural networks affect the synthesis process by correcting the heuristic value of the rules in the sub-DSL based on the output of heads:

19

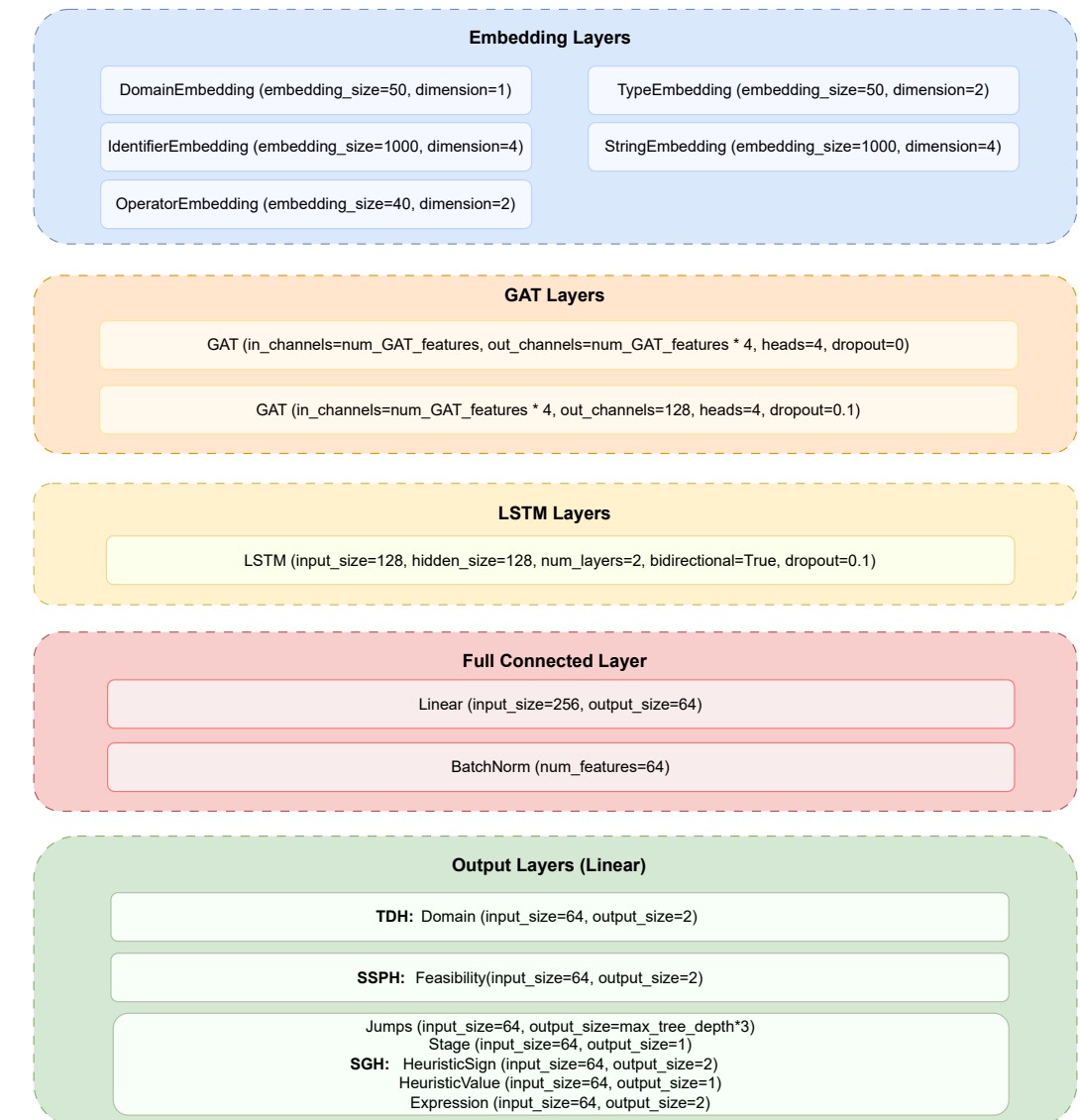

Figure 8: Layer architecture of neural networks in DSNN. Each neural network consists of embedding layers for domains, types, identifiers, strings, and operators, followed by GAT layers for tree feature extraction. LSTM layers provide sequential modeling for programs, with fully connected layers combining the outputs. Various output layers handle domain identification for task detection, feasibility judgment for search space pruning, tree jumps, stage prediction, heuristic constraint (sign and value), and constraint on the type of rule's head (expression or terminal) for search guidance.

**Task Detection Head (TDH)**: If the output indicates that the partial program does not belong to the synthesis task that the DSL can handle, any rule application on this partial program will receive an additional negative bonus on the heuristic value. For example, $\mathbf{h}[i] = \mathbf{h}[i] - |\mathbf{h}[i]| - 10$.

**Search Space Prune Head (SSPH)**: If the TDH output indicates that the partial program falls within the DSL and the SSPH output considers the partial program infeasible, any rule application on this partial program will receive an additional negative bonus on the heuristic value. For example, $\mathbf{h}[i] = \mathbf{h}[i] - |\mathbf{h}[i]| - 10$.

**Search Guidance Head (SGH)**: If the TDH output indicates that the partial program falls within the DSL and the SSPH output indicates that the partial program is promising for synthesis into a complete program, then if the features of the output rule match certain rules (logical values must be equal, and numerical values must fall within a $\pm 10\%$ range), the heuristic value when applying these rules will receive a positive bonus. For example, $\mathbf{h}[i] = \mathbf{h}[i] + |\mathbf{h}[i]|$. Otherwise, it will receive a negative bonus: $\mathbf{h}[i] = \mathbf{h}[i] - |\mathbf{h}[i]| - 10$.

### B.6.2 MULTI-DSL SYNTHESIS

The situation when multiple DSLs cooperate is similar to that of a single DSL. The difference is that for the same partial program, if at least one DSNN determines that the partial program belongs to the domain of its DSL, the DSNN bound to other DSLs, which believes that the partial program does not belong to its own domain, cannot interfere with the program synthesis at this step (as shown by signal $c$ in Figure 5).

## C EXPERIMENT

The purpose of the experiment is to explore whether CoL+NNFC DSL has advantages over traditional DSL in terms of efficiency and reliability in program synthesis from the user's perspective.

"User" refers to the user of DSL, while the developer of DSL who is proficient in specific tasks is referred to as an "expert". The experiment does not study whether CoL makes the development process easier for DSL developers, as this requires a wide range of "experts" to use COOL to develop and provide feedback. At this stage, we cannot conduct this experiment.

### C.1 USER INPUT

The user input in the experiment is COOL code containing instructions for loading the DSL (packaged as a library) and representing the task specification.

#### C.1.1 RELATIONAL TASKS

Used to test the performance of program synthesis of a single DSL:

**Code C.1: Relational Task User Input Example**

```
//load DSL for family relationship reasoning
#load(family)

//Relational reasoning questions like (50 per batch):
(Wesley) is (James)s son & (Martha) is (Wesley)s daughter &
(Hugh) is (Martha)s uncle & (Hugh) is (James)s ($relation);
...
```

#### C.1.2 SYMBOLIC TASKS

Used to test the performance of program synthesis of a single DSL:

**Code C.2: Symbolic Task User Input Example**

```
//load DSL for family relationship reasoning and symbolic
reasoning
#load(quadratic)

//Symbolic reasoning questions like (50 per batch):
$x^2 + 4*$x == 3;
...
```

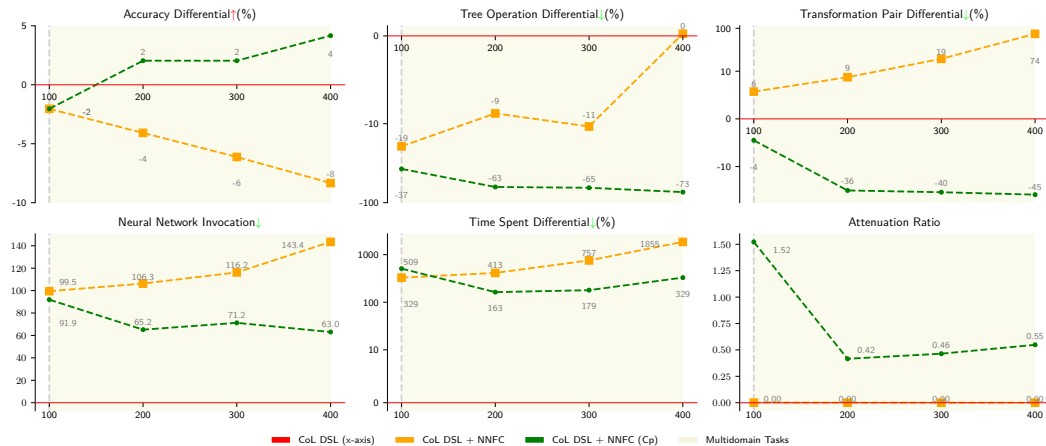

Figure 9: Dynamic performance differential to CoL DSL in multidomain tasks. The NNFC group without an inner coupling structure degrades across all 4 batches, while the group with the structure experiences degradation only in the first batch. Each batch includes 50 relational and 50 symbolic tasks, and DSNNs are continuously trained from those for tasks at difficulty level A in Figure 7.

### C.1.3 MULTI-DOMAIN TASKS

Used to test the performance of program synthesis when multiple DSLs are loaded at the same time:

> **Code C.3: Multi-Domain Task User Input Example**
>
> ```
> //load DSLs for relational reasoning and symbolic reasoning
> #load(family)
> #load(quadratic)
>
> //Symbolic reasoning questions like (50 per batch):
> $x^2 + 4*$x == 3;
> ...
> //Relational reasoning questions like (50 per batch):
> (Wesley) is (James)s son & (Martha) is (Wesley)s daughter
> & (Hugh) is (Martha)s uncle & (Hugh) is (James)s ($relation);
> ...
> ```

It should be noted that the execution of the code that does not contain the task specification is represented as a control variable in the experiment and is deducted from the final experimental results. (For example, the instructions for loading libraries)

### C.2 EXPERIMENT RESULT

## D  RULE IN CoL DSL

In addition to the heuristic vector and keywords, COOL extends the flexibility of the synthesis process by enhancing DSL rules. These enhancements are exemplified in Figure 10, which clarifies the rule introduced in Figure 1.

## E  STAGE PROGRESSION DRIVEN BY HEURISTIC VECTORS

Let $s$ denote the CoL stage, $h$ donate the heuristic value, and $n$ donate the length of CoL. A rule's heuristic vector can be mathematically represented as:

$$\mathbf{H} = \{(s_0, h_0), (s_1, h_1), \ldots, (s_n, h_n)\}, \quad n \in \mathbb{N}^+ \tag{4}$$

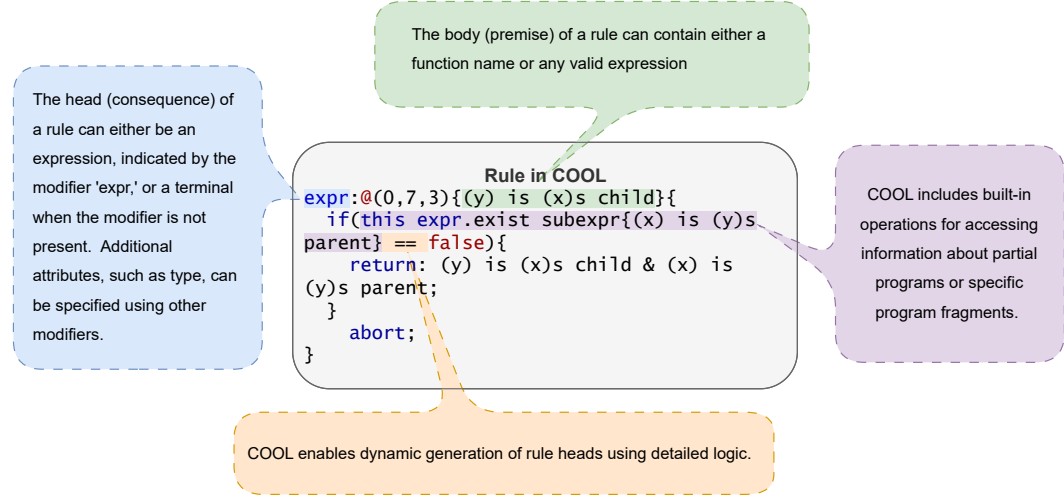

Figure 10: DSL rules in COOL. The framework allows for defining rule heads using expressions or terminals, which are enhanced with modifiers for additional attributes. Rule bodies can incorporate any valid expression or function name. Besides, COOL provides built-in operations for accessing program fragment information and facilitates dynamic rule head generation.

Upon applying a rule with heuristic vector $\mathbf{H}$, the subsequent stage, $s_{\text{next}}$, can only advance or remain the same, and the next stage should be as close to the current stage as possible:

$$\min s_{\text{next}} \quad \text{such that} \quad \exists (s_{\text{next}}, h_{\text{next}}) \in \mathbf{H} \quad \text{and} \quad s_{\text{next}} \geq s_{\text{current}} \tag{5}$$

## F    SIGNAL CLIPPER

The Clipper, as illustrated in Figure 5 (left), caps signals that do not align with the DSNN guidance to zero:

$$u_2 = \begin{cases} 0 & \text{if } u_1 > 0 \text{ and current rule doesn't align with} \\ & \text{the guidance and there exists another rule in} \\ & \text{the search space that aligns with the guidance} \\ u_1 & \text{otherwise} \end{cases} \tag{6}$$

### F.1    A* SEARCH IN PROGRAM SYNTHESIS

During the exploration phase of program synthesis, we leverage the A* algorithm to perform the heuristic search. This algorithm is renowned for its efficacy in discrete optimization tasks, utilizing heuristic guidance to navigate the search space effectively Hart et al. (1968). Each action or decision is associated with a specific cost in this context. By evaluating the cumulative cost of actions taken so far and the estimated costs of future actions, A* seeks to determine the path with the least overall cost. In our approach, heuristic values promoting forward progression are considered rewards. Therefore, we treat them as negative costs in calculations. Algorithm 1 illustrates the implementation details.

## G    IMPLEMENTATION TOOLCHAIN

To fully implement the CoL DSL and adapt it to NNFC, we choose to build COOL from the ground up rather than extending existing DSL frameworks such as Xtext Bettini (2016) or Groovy King (2020). We use C++ as the primary language to meet the execution efficiency requirements for

---

**Algorithm 1** Search Algorithm for DSL Program Synthesis

---

1: **procedure** A* SEARCH($initialPartialProgram, u_2$)
2:     $openSet \leftarrow$ priority queue containing only the initial partial program
3:     $gScore[startPartialProgram] \leftarrow 0$                                     ▷ cost from start
4:     $fScore[startPartialProgram] \leftarrow 0$
5:     **while** $openSet \neq \emptyset$ **do**
6:         $currentProgram \leftarrow openSet.\text{pop}()$     ▷ The partial program in openSet with lowest fScore value
7:         **if** $currentProgram$ is complete program **then**
8:             **return** Success
9:         **end if**
10:        **for** each $neighbor$ of $currentProgram$ **do**  ▷ Neighbor is a program directly obtained by applying a rule to the current program
11:            $tentative_gScore \leftarrow gScore[current] - u_2[neighbor]$
12:            **if** $tentative_gScore < gScore[neighbor]$ **then**
13:                $cameFrom[neighbor] \leftarrow current$
14:                $gScore[neighbor] \leftarrow tentative_gScore$
15:                $fScore[neighbor] \leftarrow gScore[neighbor] - u_2[neighbor]$
16:                **if** $neighbor \notin openSet$ **then**
17:                   $openSet.\text{add}(neighbor)$
18:                **end if**
19:            **end if**
20:        **end for**
21:     **end while**
22:     **return** Failure
23: **end procedure**

---

the numerous tree operations inherent in the DSL program synthesis process. For development efficiency, we utilize Lex Lesk & Schmidt (1975) and YACC Johnson et al. (1975) for syntax and semantic parsing, respectively. The neural network components are implemented in Python, leveraging the PyTorch library Imambi et al. (2021) to support machine learning tasks effectively. Table 8 shows the detailed code effort involved in developing the different components of COOL across various programming languages.

Table 8: Code Effort in COOL. Components of COOL are developed across different programming languages.

| Language | Lines | Components |
|---|---|---|
| C++ | 60k | framework and CoL DSL solver |
| Python | 3k | DSNN |
| Lex | 1k | syntax parser |
| YACC | 2k | semantic parsers |

## H   OPTIMIZATION STRATEGY

In practice, we observe that as the CoL length increases, the frequency of skipping stages rises. While skipping can lead to shorter synthesis paths and improved efficiency, it may cause task failures by omitting necessary stages. To manage this, we propose two strategies:

1. Gradient-Based Regulation: We employ gradient-based regulation, a widely used strategy in program synthesis Cui & Zhu (2021); Liang et al. (2018); Chaudhuri et al. (2021). By evaluating the slope or rate of change between consecutive stages, gradients help us make dynamic adjustments to synthesis paths. In our approach, we regulate skipping by applying a gradient to the heuristic values at each stage in the CoL. We encourage skipping when the heuristic gradient from one stage to the next is positive. Conversely, if the gradient is negative, we suppress skipping.

2. NNFC Regulation: Once we establish a feasible synthesis path, we can treat partial programs derived through skipping as infeasible. Then, we will utilize the feedback loop to suppress unwarranted skipping actions. However, since these partial programs might still contain feasible solutions, we need further investigation to understand and fully leverage the potential impact of this data.

In our experiments, we prioritize accuracy by suppressing skipping behavior, ensuring essential stages are included in synthesis paths.

## I    FUTURE WORK

In future work, we aim to enhance the capability of the COOL framework by exploring the implementation of CoL and NNFC in more complex scenarios, such as managing dependencies among DSL libraries and object-oriented development. We plan to facilitate community collaboration by developing more DSL libraries to expand COOL's applications. Additionally, we are interested in integrating COOL with language models. As these models evolve, ensuring ethical and accurate reasoning becomes increasingly crucial Jacovi & Goldberg (2020); Chen et al. (2022); Li et al. (2022). The COOL framework, including CoL's constraints on rule application and NNFC's structured agent interactions, helps to enhance reasoning faithfulness, preventing harmful reasoning logic. We hope our work will serve as a reliable bridge for interaction and understanding between human cognitive processes and language model reasoning.

## J    CoL DSL FOR RELATIONAL TASKS

We present only the specific code for the CoL DSL group, while the code for the DSL and DSL (Heuristic) groups, referenced in Table 3, is not displayed. This omission is because their differences from the CoL DSL group are confined to their heuristic vectors. In both the DSL and DSL (Heuristic) groups, the heuristic vectors have a dimension of 1. However, the DSL group employs a fixed heuristic value of -1, whereas the DSL (Heuristic) group utilizes variable values. The experimental codes are presented concisely, showcasing only the framework. Please refer to the attached supplementary materials for the complete content.

```
//1 Separate Relations and Genders
expr:@(9){(a) is (b)s grandson}{
    return:(a) is male &  (a) is (b)s grandchild & (b) is (a)s
    ↪  grandparent;
}
...

//2 Reason Inverse Relations
expr:@(0,7,3){(a) is (b)s grandchild}{
    if(this expr.exist subexpr{(b) is (a)s grandparent} == false){
        return: (a) is (b)s grandchild & (b) is (a)s grandparent;
    }
    abort;
}
...

//3 Reason Indirect Relations
expr:@(0,0,5){(a) is (b)s sibling}{
    placeholder:p1;
    while(this expr.find subexpr{(p1) is (a)s sibling}){
        if(this expr.exist subexpr{(p1) is (b)s sibling} == false
        ↪  && p1 != b){
return: (a) is (b)s sibling & (p1) is (b)s sibling;
        }
        p1.reset();
    }
```

```
1350        p1.reset();
1351        while(this expr.find subexpr{(p1) is (a)s parent}){
1352            if(this expr.exist subexpr{(p1) is (b)s parent} == false){
1353    return: (a) is (b)s sibling & (p1) is (b)s parent;
1354            }
1355            p1.reset();
1356        }
1357        p1.reset();
1358        while(this expr.find subexpr{(p1) is (a)s pibling}){
1359            if(this expr.exist subexpr{(p1) is (b)s pibling} ==
1360            ↪  false){
1361    return: (a) is (b)s sibling & (p1) is (b)s pibling;
1362            }
1363            p1.reset();
1364        }
1365        p1.reset();
1366        while(this expr.find subexpr{(p1) is (a)s grandparent}){
1367            if(this expr.exist subexpr{(p1) is (b)s grandparent} ==
1368            ↪  false){
1369    return: (a) is (b)s sibling & (p1) is (b)s grandparent;
1370            }
1371            p1.reset();
1372        }
1373        p1.reset();
1374        abort;
1375    }
1376    ...
1377
1378    //4 Recombine Relations and Genders, Eliminate Irrelevant
1379    ↪  Relations
1380    expr:@(0,0,0,8){(a) is (b)s ($relation)}{
1381        //immediate family
1382        placeholder:p1;
1383        while(this expr.find subexpr{(a) is (b)s grandchild}){
1384            if(this expr. exist subexpr{(a) is male}){
1385    return: $relation == "grandson";
1386            }
1387            if(this expr.exist subexpr{(a) is female}){
1388    return:$relation == "granddaughter";
1389            }
1390            p1.reset();
1391        }
1392        p1.reset();
1393        while(this expr.find subexpr{(a) is (b)s child}){
1394            if(this expr. exist subexpr{(a) is male}){
1395    return: $relation == "son";
1396            }
1397            if(this expr.exist subexpr{(a) is female}){
1398    return:$relation == "daughter";
1399            }
1400            p1.reset();
1401        }
1402        ...
1403        abort;
    }
    ...
    expr:@(0,0,0,10){a & ($b == c)}{
        return:b == c;
```

## K  CoL DSL for Symbolic Tasks

```
1404   }
1405   ...
1406
1407
1408   // Common Transformations
1410   expr:@(2,2,2,2,2){0+#a}{
1411       return:a;
1412   }
1413   expr:@(2,2,2,2,2){#a+0}{
1414       return:a;
1415   }
1416   ...
1417
1418   // 1 Expand Square Terms
1419   expr:@(5,0,0,0){(#?a + #?b)^2}{
1420       return:a^2+2*a*b+b^2;
1421   }
1422   expr:@(5,0,0,0){(#?a - #?b)^2}{
1423       return:a^2+(-2)*a*b+b^2;
1424   }
1425   expr:@(6,0,0,0){(#a*#b)^2}{
1426       return:a^2*b^2;
1427   }
1428   ...
1429
1430   // 2 Expand Bracketed Terms
1431   expr:@(0,4,0,0,0){#?a-(#?b+#?c)}{
1432       return:a-b-c;
1433   }
1434   expr:@(0,3.8,0,0,0){(#?b+#?c)*#?a}{
1435       return:b*a+c*a;
1436   }
1437   ...
1438
1439   // 3 Extract Coefficients
1440   expr:@(0,0,5,0){$x*a}{
1441       return:a*x;
1442   }
1443   expr:@(0,0,4.8,0){(immediate:a*$x)*(immediate:b*$x)}{
1444       new:tmp = a*b;
1445       return:tmp*x^2;
1446   }
1447   expr:@(0,0,4.6,0){$x*(a*$x)}{
1448       return:a*x^2;
1449   }
1450   ...
1451
1452   // 4 Re-Express Negative Coefficients
1453   expr:@(0,0,0,3.5,0){#a-$x}{
1454       placeholder:p1;
1455       placeholder:p2;
1456       if(x.exist subexpr{p1*p2}){
1457           abort;
       }
       return:a+(-1)*x;
   }
   expr:@(0,0,0,3.7,0){#a-immediate:b*$x}{
```

```
new:tmp = 0 - b;
    return:a+tmp*x;
}
...

//5 Arrange Terms in Descending Order, Combine Like Terms
expr:@(0,0,0,0,3){immediate:a*$x+immediate:b*$x}{
    new:tmp = a+b;
    return:tmp*x;
}
expr:@(0,0,0,0,2.8){a1*$x+a2*$x^2}{
    return:a2*x^2+a1*x;
}
...

//6 Convert to Standard Form
expr:@(0,0,0,0,0,2.5){a*$x^2+b*x == #d}{
    return: a*$x^2+b*x + 0 == d;

}
expr:@(0,0,0,0,0,2.5){b*$x == $d}{

    if(d.exist subexpr{x^2}){
        return: 0*x^2 + b*x + 0 == d;
    }else {
        abort;
    }
}
expr:@(0,0,0,0,0,-4){$a==$b}{
    return:b==a;
}
...

//7 Apply Solution Formula
@(0,0,0,0,0,0,0,10){a*$x^2+b*x+c==0}{
    if(b^2-4*a*c<0){
        x="null";
    }
    else {
        new:x1=(-b+(b^2-4*a*c)^0.5)/(2*a);
        new:x2=(-b-(b^2-4*a*c)^0.5)/(2*a);
        x={x1,x2};
    }
};
```

## L  RELATIONAL TASKS AT DIFFICULTY LEVEL A

```
#load(family) // Load the CoL DSL library for Relational Tasks
new:relation = "";
// [Francisco]'s brother, [Wesley], recently got elected as a
↪  senator. [Lena] was unhappy with her son, [Charles], and his
↪  grades. She enlisted a tutor to help him. [Wesley] decided to
↪  give his son [Charles], for his birthday, the latest version
↪  of Apple watch.
// Ans: (Francisco) is (Lena)s brother
new:Lena = "Lena";
new:Charles = "Charles";
new:Wesley = "Wesley";
```

```
1512  new:Francisco = "Francisco";
1513  (Charles) is (Lena)s son & (Wesley) is (Charles)s father &
1514  ↪   (Francisco) is (Wesley)s brother & (Francisco) is (Lena)s
1515  ↪   ($relation);
1516  relation-->"#FILE(SCREEN)";
1517
1518  // [Clarence] woke up and said hello to his wife, [Juanita].
1519  ↪   [Lynn] went shopping with her daughter [Felicia]. [Felicia]'s
1520  ↪   sister [Juanita] was too busy to join them.
1521  // Ans: (Lynn) is (Clarence)s mother-in-law
1522  new:Clarence = "Clarence";
1523  new:Juanita = "Juanita";
1524  new:Felicia = "Felicia";
1525  new:Lynn = "Lynn";
1526  (Juanita) is (Clarence)s wife & (Felicia) is (Juanita)s sister &
1527  ↪   (Lynn) is (Felicia)s mother & (Lynn) is (Clarence)s
1528  ↪   ($relation);
1529  relation-->"#FILE(SCREEN)";
      ...
```

## M  RELATIONAL TASKS AT DIFFICULTY LEVEL B

```
1536  #load(family) // Load the CoL DSL library for Relational Tasks
1537  new:relation = "";
1538  // [Antonio] was happy that his son [Bernardo] was doing well in
1539  ↪   college. [Dorothy] is a woman with a sister named [Tracy].
1540  ↪   [Dorothy] and her son [Roberto] went to the zoo and then out
1541  ↪   to dinner yesterday. [Tracy] and her son [Bernardo] had lunch
1542  ↪   together at a local Chinese restaurant.
1543  // Ans: (Roberto) is (Antonio)s nephew
1544  new:Antonio = "Antonio";
1545  new:Bernardo = "Bernardo";
1546  new:Tracy = "Tracy";
1547  new:Dorothy = "Dorothy";
1548  new:Roberto = "Roberto";
1549  (Bernardo) is (Antonio)s son & (Tracy) is (Bernardo)s mother &
1550  ↪   (Dorothy) is (Tracy)s sister & (Roberto) is (Dorothy)s son &
1551  ↪   (Roberto) is (Antonio)s ($relation);
1552  relation-->"#FILE(SCREEN)";
1553
1554  // [Bernardo] and his brother [Bobby] were rough-housing. [Tracy],
1555  ↪   [Bobby]'s mother, called from the other room and told them to
1556  ↪   play nice. [Aaron] took his brother [Bernardo] out to get
1557  ↪   drinks after a long work week. [Tracy] has a son called
1558  ↪   [Bobby]. Each day they go to the park after school. ans:
1559  ↪   (Bobby) is (Aaron)s brother
1560  new:Aaron = "Aaron";
1561  new:Bernardo = "Bernardo";
1562  new:Bobby = "Bobby";
1563  new:Tracy = "Tracy";
1564  (Bernardo) is (Aaron)s brother & (Bobby) is (Bernardo)s brother &
1565  ↪   (Tracy) is (Bobby)s mother & (Bobby) is (Tracy)s son & (Bobby)
      ↪   is (Aaron)s ($relation);
      relation-->"#FILE(SCREEN)";
      ...
```

# N   SYMBOLIC TASKS AT DIFFICULTY LEVEL A

```
#load(quadratic) // Load the CoL DSL library for Symbolic Tasks
new:x = 1;
6*$x^2 == 3*x - 7;
x-->"#FILE(SCREEN)";
($x - 6)*(x + 3) == x;
x-->"#FILE(SCREEN)";
...
```

# O   SYMBOLIC TASKS AT DIFFICULTY LEVEL B

```
#load(quadratic) // Load the CoL DSL library for Symbolic Tasks
new:x = 1;
$x*($x + 11) == 16*($x + 22);
x-->"#FILE(SCREEN)";
$x*(36*$x + 50) - 11*(19 - 30*$x) == $x^2;
x-->"#FILE(SCREEN)";
...
```

# P   MULTIDOMAIN TASKS

```
#load(quadratic) // Load the CoL DSL library for Symbolic Tasks
#load(family) // Load the CoL DSL library for Relational Tasks
new:x = 1;
$x^2 - 4*$x == 6;
x --> "#FILE(SCREEN)";
...
new:relation = "";
// [Dolores] and her husband [Don] went on a trip to the
↪  Netherlands last year. [Joshua] has been a lovely father of
↪  [Don] and has a wife named [Lynn] who is always there for him.
// Ans: (Dolores) is (Lynn)s daughter-in-law
new:Lynn = "Lynn";
new:Joshua = "Joshua";
new:Don = "Don";
new:Dolores = "Dolores";
(Joshua) is (Lynn)s husband & (Don) is (Joshua)s son & (Dolores)
↪  is (Don)s wife & (Dolores) is (Lynn)s ($relation);
relation-->"#FILE(SCREEN)";
...
```

# Q   COOL INTERMEDIATE REPRESENTATION

The intermediate representation of COOL is Three-Address Code.

```
"codeTable": [
    {
        "boundtfdomain": "",
        "grounded": false,
        "operand1": {
            "argName": "x",
            "argType": "identifier",
            "changeable": 1,
            "className": "",
            "isClass": 0
        },
```

```
            "operand2": {
                "argName": "2",
                "argType": "number",
                "changeable": 0,
                "className": "",
                "isClass": 0
            },
            "operator": {
                "argName": "^",
                "argType": "other"
            },
            "result": {
                "argName": "1418.4",
                "argType": "identifier",
                "changeable": 1,
                "className": "",
                "isClass": 0
            },
            "root": false
        },
        ...
]
```

## R  RELATED WORK

**Neural Search Optimization:** Neural networks are key for optimizing search in program synthesis. Projects like Kalyan et al. (2018); Zhang et al. (2023) and Li et al. (2024) use neural networks to provide oracle-like guidance, while Neo Feng et al. (2018), Flashmeta Polozov & Gulwani (2015), and Concord Chen et al. (2020) prune search spaces with infeasible partial programs. COOL employs both strategies to enhance efficiency.

**Multi-step Program Synthesis:** Chain-of-Thought (CoT) Wei et al. (2022) enhances LLMs by breaking tasks into subtasks. Projects like Zhou et al. (2022); Shi et al. (2023b) and Zheng et al. (2023) use this in program synthesis. Compared to CoT, which directly decomposes tasks, CoL does so indirectly by constraining rule applications.

**Reinforcement Learning:** Reinforcement learning improves neural agents in program synthesis through feedback, as seen in Eberhardinger et al. (2023); Liu et al. (2024); Bunel et al. (2018), Concord Chen et al. (2020), and Quiet-STaR Zelikman et al. (2024). NNFC similarly refines control flow but serves an auxiliary role for programmer strategies in synthesis rather than dominating it.

