# OpenReview forum: "COOL: Efficient and Reliable Chain-Oriented Objective Logic with Neural Networks Feedback Control for Program Synthesis"
_ICLR.cc/2025/Conference — ICLR 2025 Conference Withdrawn Submission_

### Official Review · Reviewer_7PzC · 2024-10-31

**Soundness:** 1
**Presentation:** 1
**Contribution:** 2
**Rating:** 3
**Confidence:** 3

**Summary:**

I do not feel I have a sufficient understanding of this paper to summarize it in detail. See the Weaknesses section for details on this. My best attempt to summarize this paper follows.

COOL is a system for neurosymbolic program synthesis, in which programs are generated incrementally in a DSL annotated with specific kinds of assertions and guides that allow for specific guidance of the neural network that drives the search. This leads to significant pruning, improving accuracy while reducing the size of the search space.

**Strengths:**

The results seem very impressive, showing both an improvement in performance and a reduction in computational cost.

**Weaknesses:**

This paper provides insufficient details and precision for a reader to understand the nature of the algorithm. The following questions remain unanswered by the introduction, figures, and methods section: What is the artifact a user of this system needs to provide? The baseline is to provide a DSL, what additional information needs to be provided to use COOL?

The details on the amount of effort a user needs to complete to make a DSL in COOL are crucial because without this information, it is unclear how much of the benefits shown in the results section are downstream of the user providing more scaffolding and thus making the task easier.

The figures do not help clarify this to a reader as it is not clear whether they represent the user-provided activity diagram, an execution trace for an activity diagram, or a schematic explanation of how the algorithm functions. It is also unclear what heuristic vectors are, and when logic jumps / abort commands should be used, or even if these are being written by a programmer or the output of some neural network.

Additionally, it is never specified how the DSNN network is trained. How is the data gathered for the DSNN? Is it trained via supervised learning, some kind of wake/sleep loop, or reinforcement learning? What does it mean for the DSNN to be pretrained?

**Questions:**

See Weaknesses section for a series of questions regarding this system.

---

> ### Author Response · Authors · 2024-11-14
> **Clarifying Confusion and Applying for a Detailed Review**
>
> Thank you for your review. The content of the paper is a bit complex, and let me first introduce some concepts of DSL program synthesis and its relationship with neural network training (I will also organize it separately in the appendix):
> In DSL, a complete program is synthesized by applying rules to partial programs. Each partial program (P) and the corresponding rules (R) form a transformation pair (P, R). When a rule is applied, it modifies the syntax tree of the partial program through what we refer to as a tree operation. The complete synthesis process consists of a series of transformation pairs connected by tree operations, represented as [(P0, R0), (P1, R1), ... (complete P/failure/other)], which is known as a synthesis path or trajectory.
>
> The essence of the DSL synthesis process is a search problem focused on efficiently constructing this path. To learn from earlier experience, the neural network leverages these transformation pairs (P, R) on synthesis paths for supervised learning, with P used as the input and R as the label.
>
> ```
> What is the artifact a user of this system needs to provide?
> ```
> The "expert" familiar with the domain-specific problems writes the DSL and general users directly use the DSL to solve problems. Only the users' programs are regarded as inputs. In other words, you can think DSL is a kind of library, and the library developed with CoL + NNFC is better at efficiency and reliability. The experiments are focused on proving CoL + NNFC DSLs perform better than general DSLs. (but I can add a section in the appendix to explain the input-output in detail if necessary.). Therefore, users who use (CoL + NNFC) DSLs needn't provide anything else but synthesis tasks.
>
> ```
> The details on the amount of effort a user needs to complete to make a DSL in COOL are crucial because without this information, it is unclear how much of the benefits shown in the results section are downstream of the user providing more scaffolding and thus making the task easier.
> ```
> Same as the above question. This paper focuses on enhancing DSLs with CoL and NNFC so that they can tackle more complex tasks and show higher efficiency and reliability in challenging scenarios.
>
>
> ```
> The figures do not help clarify this to a reader as it is not clear whether they represent the user-provided activity diagram, an execution trace for an activity diagram, or a schematic explanation of how the algorithm functions. It is also unclear what heuristic vectors are, and when logic jumps / abort commands should be used, or even if these are being written by a programmer or the output of some neural network.
> ```
> In the Introduction, the paper said, "we introduce the Chain-of-Logic (CoL), which integrates the functions of the activity diagram to enable fine-grained control" It means the CoL enables programmers to control the synthesis flow just like writing an activity diagram: Programmer can use heuristic vectors to decompose the synthesis process into sequential stages, each stage has corresponding rules can be applied. Then, use return, logicjump, and abort to organize the control flow between stages. As for how a heuristic vector specifies the applicable stage(s) of a function, the paper puts up an example in Section 2.1 :"a rule with the heuristic vector (0,7,3) is applicable in stages 2 and 3 with heuristic values of 7 and 3, respectively.". Besides, Figure 3 demonstrates how heuristic vectors and keywords are utilized to build up the control flow. It is worth noticing that all these are written by the "expert" rather than users or neural networks.
>
> ```
> How is the data gathered for the DSNN?
> ```
>
> As mentioned in the beginning, the training data is composed of transformation pairs in synthesis paths. Specifically, a data set for training a DSNN for DSL "A" is as follows: (P R) of other DSLs for training Task Detection Head (TDH), (P R) of DSL "A" but in infeasible synthesis paths for training Search Space Prune Head (SSPH), and (P R) of DSL "A" and in feasible synthesis pathes for training Search Guidance Head (SGH). I will provide an overview of the export process in the Appendix if necessary.
>
>
> ```
> How is the DSNN trained?
> ```
> Besides supervised learning, as a new DSL is created, a new NNFC feedback loop is created automatically to assist CoL DSL synthesis. And the DSNN within it will be trained from scratch and continuously trained to improve itself (See captions of Figure 6 and 7). Additionally, DSNNs apply a filtering structure to eliminate mispredictions and ensure overall reliability.
>
> If you have any other questions, please let me know, and I will answer them as soon as possible.

---

> > ### Comment · Reviewer_7PzC · 2024-11-16
> > **Response to author's response**
> >
> > > Each partial program (P) and the corresponding rules (R) form a transformation pair (P, R). When a rule is applied, it modifies the syntax tree of the partial program through what we refer to as a tree operation. The complete synthesis process consists of a series of transformation pairs connected by tree operations, represented as [(P0, R0), (P1, R1), ... (complete P/failure/other)], which is known as a synthesis path or trajectory… To learn from earlier experience, the neural network leverages these transformation pairs (P, R) on synthesis paths for supervised learning, with P used as the input and R as the label
> >
> > This absolutely needs to be explained directly in the text of the paper. There are multiple techniques for DSL program synthesis and only some (generally top-down ones) involve applying rules to partial programs in a sequential manner as described above. Additionally, the fact that the neural network is directly predicting R from P is not shared in many neural guided top-down synthesis algorithms.
> >
> > > The "expert" familiar with the domain-specific problems writes the DSL and general users directly use the DSL to solve problems. Only the users' programs are regarded as inputs. In other words, you can think DSL is a kind of library, and the library developed with CoL + NNFC is better at efficiency and reliability. The experiments are focused on proving CoL + NNFC DSLs perform better than general DSLs. (but I can add a section in the appendix to explain the input-output in detail if necessary.). Therefore, users who use (CoL + NNFC) DSLs needn't provide anything else but synthesis tasks… Same as the above question. This paper focuses on enhancing DSLs with CoL and NNFC so that they can tackle more complex tasks and show higher efficiency and reliability in challenging scenarios.
> >
> > A list of the specific information that a user needs to provide in addition to a basic BNF that you’d find in a standard DSL is absolutely crucial in being able to evaluate the practical utility of this effect -- otherwise there is  a concern that the synthesis algorithm involves getting an expert to write a domain-specific synthesis algorithm.
> >
> > > In the Introduction, the paper said, "we introduce the Chain-of-Logic (CoL), which integrates the functions of the activity diagram to enable fine-grained control" It means the CoL enables programmers to control the synthesis flow just like writing an activity diagram: Programmer can use heuristic vectors to decompose the synthesis process into sequential stages, each stage has corresponding rules can be applied. Then, use return, logicjump, and abort to organize the control flow between stages. As for how a heuristic vector specifies the applicable stage(s) of a function, the paper puts up an example in Section 2.1 :"a rule with the heuristic vector (0,7,3) is applicable in stages 2 and 3 with heuristic values of 7 and 3, respectively.". Besides, Figure 3 demonstrates how heuristic vectors and keywords are utilized to build up the control flow. It is worth noticing that all these are written by the "expert" rather than users or neural networks.
> >
> > I am still confused after reading this explanation as to what exactly “heuristic values” are, and how exactly control flow works. Is there a language for expressing this control flow? If so, is it specified somewhere in the paper?
> >
> > > As mentioned in the beginning, the training data is composed of transformation pairs in synthesis paths. Specifically, a data set for training a DSNN for DSL "A" is as follows: (P R) of other DSLs for training Task Detection Head (TDH), (P R) of DSL "A" but in infeasible synthesis paths for training Search Space Prune Head (SSPH), and (P R) of DSL "A" and in feasible synthesis pathes for training Search Guidance Head (SGH). I will provide an overview of the export process in the Appendix if necessary.
> >
> > The process of gathering feasible and infeasible paths for this dataset is of great importance; as this is a task of turning a non-supervised learning problem into a supervised learning problem, and as such, the exact distributional qualities of this data are significant to how the model training works.

---

> > > ### Author Response · Authors · 2024-11-16
> > >
> > > ```
> > > This absolutely needs to be explained directly in the text of the paper. There are multiple techniques for DSL program synthesis and only some (generally top-down ones) involve applying rules to partial programs in a sequential manner as described above. Additionally, the fact that the neural network is directly predicting R from P is not shared in many neural guided top-down synthesis algorithms.
> > > ```
> > > Sure, I will elaborate on this. "neural networks learn from (P, R)" is an easy-to-understand but not rigorous statement. To be honest, neural networks do not generate a rule directly. Instead, they generate the features to specify rule(s). And, this is what Search Guidance Head (SGH) does. For Task Detection Head (TDH) and Search Space Prune Head (SSPH), they make decision directly on P.
> > >
> > > ```
> > > A list of the specific information that a user needs to provide in addition to a basic BNF that you’d find in a standard DSL is absolutely crucial in being able to evaluate the practical utility of this effect -- otherwise there is a concern that the synthesis algorithm involves getting an expert to write a domain-specific synthesis algorithm.
> > > ```
> > > I will make it clear. The COOL adopts a parenthesis-free syntax (not traditional BNF) for easy usage and expressiveness. DSLs are libraries not written by "users"(see https://coologic.org/COOL_Syntax/Syntax_Manual.pdf). "Users" just need to load the library and write the specification, and we monitor specific metrics in the reasoning process to evaluate DSLs and their variants:
> > > What a user provides:
> > >
> > > #load(quadratic) // Load the CoL DSL library for Symbolic Tasks
> > > new:x = 1;//initialize the variable
> > > $x^2 - 4*$x == 6; // the detailed problem user wants to solve (experiment tasks)
> > > ... // more problems
> > >
> > > "Experts" don't have to write the synthesis algorithm; they just write the DSL based on their expertise, and that's the starting point of designing COOL.
> > >
> > > ```
> > > I am still confused after reading this explanation as to what exactly “heuristic values” are, and how exactly control flow works. Is there a language for expressing this control flow? If so, is it specified somewhere in the paper?
> > > ```
> > > The "heuristic vectors(vectors at the beginning of rule/function declarations )" directly organize the control flow. Here is a more formal version of how heuristic vector works ( please refer to the DSL code in Appendix H and I):
> > > The original program synthesis process is divided into multiple sequential stages. In each stage, the intermediate form of the program generated from the earlier stage is regarded as this stage's input, which is handled by the sub-DSL of this stage. The rules of each sub-DSL are selected from the set of rules of the original DSL, and the selection is based on the fact that the specific bit in the heuristic vector of the rule is not zero.
> > >
> > > As for heuristic values, they are factors that influence the choice of rules during exploration. For example, given the synthesis process is a path of transformation pairs, we adopt A* as the synthesis algorithm and regard heuristic values as rewards, so the rules applicable in this stage with bigger heuristic values are prioritized and frequently called.
> > >
> > > However, there is no one-size-fits-all synthesis algorithm for different problems. The implementation interface of the synthesis algorithm based on heuristic value will be open to "experts" who write DSLs.
> > >
> > > ```
> > > The process of gathering feasible and infeasible paths for this dataset is of great importance, as this is a task of turning a non-supervised learning problem into a supervised learning problem, and as such, the exact distributional qualities of this data are significant to how the model training works.
> > > ```
> > > I will detail the data collection and augmentation process.

---

> > > > ### Author Response · Authors · 2024-11-16
> > > >
> > > > The first draft of this paper is more than 50 pages. In order to compress the content, we refer to papers of complex projects (such as pytorch). These papers do not discuss a lot of details in the main text by showing innovations and proving the advancedness of the technology through strong experimental data. The core innovations of this paper are CoL (Chain-of-Logic) and NNFC (Neural Network Feedback Control). These two structures are innovations at the level of DSL program synthesis framework. The neural network belongs to the module of the feedback loop, and the synthesis algorithm belongs to the module of the forward path. They belong to the second level under CoL and NNFC, so they have priority in the narrative. Therefore, when the text content is fully compressed, these contents cannot be described in detail in the main text. We hope you can understand our approach.
> > > >
> > > > We will provide more detailed examples, more detailed descriptions of the synthesis algorithm, and more detailed neural network life cycle in the appendix, and add convenient clickable jumps to the appendix in the main text, so that readers can easily understand the technology in the text from an abstract level and specific implementation. If you have any questions, please feel free to ask me; if you recognize the innovation of this paper and its contribution to DSL program synthesis, please allow this paper to be passed, which will promote the research of COOL. Thank you.

---

### Official Review · Reviewer_gk7j · 2024-11-02

**Soundness:** 2
**Presentation:** 2
**Contribution:** 2
**Rating:** 5
**Confidence:** 2

**Summary:**

The paper introduces a neuro-symbolic program synthesis method that combines rule-based logic with neural feedback, termed Chain-Oriented Objective Logic (COOL). This technique allows fine-grained control over search processes within DSL-induced program spaces. Modularity is achieved through a component called Chain of Logic (CoL), inspired by the activity diagram model from software engineering and system control. CoL structures complex rule applications by assigning heuristic values to rules, enabling more efficient search strategies. Selecting specific rules based on heuristic values at particular stages (e.g., returning, jumping, or terminating synthesis branches) enhances search efficiency by aligning rule applications with promising paths, akin to human problem-solving approaches.

The paper also introduces Neural Network Feedback Control (NNFC), which uses multiple neural networks in series with specific heads for task detection, search pruning, and guidance. This prioritizes rules and tasks, allowing the synthesis process to bypass infeasible paths early, thereby improving the quality of synthesized outputs.

Experiments were conducted on static and dynamic benchmarks, categorized by difficulty level. The results validate both the general applicability of the approach and impressive improvements in accuracy and search efficiency under different task demands. A detailed ablation study clarifies the contribution of each component.

**Strengths:**

1. The ideas introduced in the paper are interesting. The Chain of Logic structure mirrors human problem-solving steps, allowing for more intuitive management of the program synthesis process within the DSL. This hierarchical structuring likely enhances interpretability.
2. The paper provides some experimental validation, with ablation studies showing the contribution of each component.

**Weaknesses:**

1. The CoL framework is highly domain-specific, and it is unclear how adaptable it would be for tasks outside its designed domain. The reliance on activity diagrams suggests that applying CoL in other fields may require substantial modification, limiting generalizability.
2. The description of NNFC is somewhat opaque, especially regarding how incorrect neural network predictions are “suppressed through filtering.”
3. There is insufficient detail about the design choices behind the Domain-Specific Neural Networks (DSNNs), such as how they are trained and what training data or ground truth is used, especially when handling partial programs.

**Questions:**

1. The process by which DSNNs generate error signals from partial programs is not clearly explained. What constitutes a “correct” path in the training process? Is there a universal ground truth, or does it vary by domain? Also how much data does each DSNN require to perform effectively?
2. Did the dynamic experiments include any resource constraints (e.g., time or computational power limits)?
3. I did not understand what “multidomain tasks” are and what is their role in the experiments. Can you clarify please?

---

> ### Author Response · Authors · 2024-11-17
>
> Thank you for your positive comments on the paper's innovations.
>
>
> ```
> The CoL framework is highly domain-specific, and it is unclear how adaptable it would be for tasks outside its designed domain. The reliance on activity diagrams suggests that applying CoL in other fields may require substantial modification, limiting generalizability.
> ```
> CoL framework is generalizable but a specific CoL / NNFC implementation is highly domain-specific. Specifically, CoL is implemented through heuristic vectors and keywords, which are closely bound to the functions(rules) in DSLs. You can regard a CoL/NNFC implementation as a part of the improved DSL.
>
> ```
> The description of NNFC is somewhat opaque, especially regarding how incorrect neural network predictions are “suppressed through filtering.”
> ```
>
>
> Detailed description of Figure 4 (left):
>
> Forward path (CoL): Heuristic values guide rule application in the synthesis process.
>
> (How a DSL works: A complete program is synthesized in DSL by applying rules to partial programs. Each partial program (P) and the corresponding rules (R) form a transformation pair (P, R). When a rule is applied, it modifies the syntax tree of the partial program through what we refer to as a tree operation. The complete synthesis process consists of a series of transformation pairs connected by tree operations, represented as [(P0, R0), (P1, R1), ... (complete P(feasible path)/failure(infeasible path)/other )], which is known as a synthesis path or trajectory.
>
> How heuristic values work: We use heuristic values to influence the choice of rules during exploration. For example, given the synthesis process is a path of transformation pairs, we adopt A* as the synthesis algorithm and regard heuristic values as rewards, so the rules applicable in this stage with bigger heuristic values are prioritized and frequently called.)
>
>
>
> Feedback path (NNFC, which depends on CoL and influences/corrects the heuristic values in CoL):
>
>  Its input (from a perspective of feedback) consists of transformation pairs from synthesis paths:  Task Detection Head (TDH) needs synthesis paths of other DSL(s) and built-in function invocations to train; Search Space Prune Head (SSPH) needs infeasible synthesis paths of current DSL; Search Guidance Head (SGH) needs feasible synthesis paths of current DSL.
>
> Its output generates signals to adjust CoL's heuristic value. This adjustment is determined by the outputs from the Task Detection Head (TDH), Search Space Prune Head (SSPH), and Search Guidance Head (SGH). Specifically, the NNFC outputs correct the heuristic value in the following ways:
>
> TDH --(corresponding to) -- domain(detailed in Table 7 of the Appendix, same as below): The relevance of task domains to DSNN. If the current partial program does not align with the DSL's field, the heuristic value gets a negative gain.
>
> SSPH -- feasibility: The viability of synthesizing the complete program. If the current partial program cannot produce the complete program, the heuristic value gets a negative gain.
>
> SGH -- other output features in Table 7: Characteristics that rules for partial programs should meet. Consistency provides a positive gain for heuristic value, while inconsistency results in a negative gain. I can expand on these details in the Appendix.
>
>
> Detailed description of Figure 4 (right):
>
> The neural networks in a DSNN are connected in series (For detailed input features of each network, please refer to Table 6 ). When the output of an upstream neural network is unreliable, the output of the downstream neural network that relies on its output as input will become vulnerable. When the output of the upstream neural network is unreliable, the output of the downstream neural network that uses its output as input will become vulnerable. Then compare the same output features in the upstream and downstream neural networks for cross-validation. If these features differ greatly (manually set thresholds), it means that the prediction is unreliable and can be eliminated. (I will elaborate on this part in the appendix)
>
> ```
> There is insufficient detail about the design choices behind the Domain-Specific Neural Networks (DSNNs), such as how they are trained and what training data or ground truth is used, especially when handling partial programs.
>
> The process by which DSNNs generate error signals from partial programs is not clearly explained. What constitutes a “correct” path in the training process? Is there a universal ground truth, or does it vary by domain? Also how much data does each DSNN require to perform effectively?
>
> ```
> Same as above (I will make up for this part in the appendix)

---

> > ### Author Response · Authors · 2024-11-17
> >
> > ```
> > Did the dynamic experiments include any resource constraints (e.g., time or computational power limits)?
> > ```
> > Yes. Each partial program must be completed with at most 1000 transformation pairs, though this may exceed
> > 1000 if additional tasks are generated during synthesis. (the footnote on page 6) This is a constraint on memory usage because transformation pairs consume a lot of memory.
> >
> > ```
> > What “multidomain tasks” are and what is their role in the experiments.
> > ```
> >
> > This corresponds to the situation where multiple DSLs collaborate to process tasks, which belongs to the study of DSL modularization (the main difficulty lies in accurately identifying tasks in their own domain when multiple DSLs collaborate and avoiding processing tasks that do not belong to their own domain):
> >
> > #load(quadratic) // Load the CoL DSL library for Symbolic Tasks
> >
> > #load(family) // Load the CoL DSL library for Relational Tasks
> >
> > $xˆ2 - 4*$x == 6; // Type A task ...
> >
> > (Joshua) is (Lynn)s husband & (Don) is (Joshua)s son & (Dolores) is (Don)s wife & (Dolores) is (Lynn)s ($relation);  // Type B task...
> >
> > In the experiment, two DSLs are called simultaneously for synthesis tasks to verify that NNFC enables DSL to acquire additional ability to distinguish its own tasks from other tasks.
> >
> > If you have any more questions, please let me know. And if you think our innovations can promote research in the field of DSL program synthesis, you can give a higher score for this paper. Thank you.

---

> > > ### Comment · Reviewer_gk7j · 2024-11-25
> > >
> > > Thanks for your response. I do not have any further questions at the moment.

---

### Official Review · Reviewer_7Vwc · 2024-11-04

**Soundness:** 1
**Presentation:** 1
**Contribution:** 1
**Rating:** 1
**Confidence:** 4

**Summary:**

This paper describes a form of program synthesis called COOL and applies it to the CLUTTR dataset (logical family relation problems embedded in natural language) and a toy quadratic equation dataset. COOL is a combination of a "Chain of Logic" (CoL) DSL and a neural network feedback control system (NFCC) for guiding search. CoL is a framework for sketching templates of useful DSL rule applications to guide the problem solving process. For example, for the CLUTTR dataset, the CoL DSL contains rules "Separate relations and genders", "Reason inverse relations", "Reason Indirect Relations", "Recombine relations and genders, eliminate irrelevant info", while the CoL DSL for quadratic equations contains rules such as 0 + a = a, (a + b) ^ 2 = a^2 + 2ab + b^2. NFCC is a complicated system that uses neural networks to guide and control the synthesis process. Program search is conducted with A* using heuristics I believe from the neural network. The authors show that COOL achieves 100% accuracy, uses fewer tree operations, and spends much less time solving compared to the baseline of a normal DSL without NNFC.

**Strengths:**

The paper is very thorough and complex. There is a lot of detail and technical content. I have never seen a control system like that used for the neural network feedback control and it seems like a very novel way of synthesizing programs. While I don't fully understand it, the authors analysis of inner coupling as a key factor to the improved synthesis quality of COOL seems interesting.

**Weaknesses:**

I find this paper very hard to understand.

The method is very complicated, with a lot of moving parts. The method is never described end-to-end in a coherent way — instead, bits of detail are mentioned throughout the paper, but I never get the sense that I fully understand what is going on. Some questions that I cannot answer from reading the paper: how are the neural networks trained? how are the heuristics computed? what is the input and output for a given synthesis problem? what are tree operations? One of the datasets, the quadratic equation dataset, is only described at a high level — I don't even know what the problems look like exactly.

I would recommend writing the paper in a more linear order. Even if there is not space in the main text, the appendix can contain a full description of the (1) problem input and output, (2) overview of algorithm that transforms input to output prediction, (3) description of how your techniques plug in to the algorithm at different parts. You can also include more detailed definitions for things like tree operations, the DSL rule definitions, etc.

Overall, I am baffled at this paper. I am not foreign to program synthesis, yet most of the content of this paper is beyond me.

To summarize, the technique is extremely complicated and not adequately explained. This is the main weakness of the paper.

Beyond this, the evaluation is not very convincing. This is in large part due to the poor explanation — as stated earlier, I don't have a full understanding of what the input and output for each problem is, or even how search is conducted. So it's hard to know what to make of the accuracy metrics or time spent. "tree operation" and "transformation pair" are not defined either.

**Questions:**

See weaknesses section for questions

---

> ### Author Response · Authors · 2024-11-14
> **Clarifying Confusion and Applying for a Detailed Review**
>
> Thank you for taking the time to review the paper. I acknowledge its interdisciplinary nature and complexity, which we must tackle thoughtfully.
>
>
> It seems your main concern is with the narrative style of the paper, which you feel is not sufficiently end-to-end or linear. This perception arises from the reality that the technology discussed is not conducive to a straightforward end-to-end discussion. Specifically, I believe your confusion primarily stems from the second section, "Method." This section outlines the entire control system in a progressive manner: it first introduces the forward path, which details CoL (Chain-of-Logic)'s control of the synthesis process, and then describes the feedback loop where NNFC(Neural Network Feedback Control) corrects CoL. This structure aligns with the logical framework of the control system and reflects a more effective narrative flow, as we discussed. It's important to note that a sequential narrative approach is more suited for unidirectional systems and may not adequately capture the intricacies of the feedback structure we are presenting.
>
> Since this narrative approach differs from your usual style, allow me to clarify the technical framework:
> First, the paper presents the CoL DSL+NNFC synthesis system as a cohesive unit. Figure 1 clearly outlines its input-output relationship; the input is a partial program, and the output is a complete program.
> In the "Method" section, the structure of the CoL DSL+NNFC synthesis system is introduced sequentially, beginning with the forward path followed by the feedback loop:
> Forward path: CoL DSL, where the input and output remain as a partial program and a complete program, respectively. Here, heuristic values guide rule application.
> Feedback path: NNFC (Neural Network Feedback Control), which depends on CoL. Its input consists of training data (transformation pairs) derived from partial programs during the synthesis process, while its output generates signals to adjust CoL's heuristic value. This adjustment is determined by the outputs from the Task Detection Head (TDH), Search Space Prune Head (SSPH), and Search Guidance Head (SGH) as described in section 2.2. Specifically, the NNFC outputs correct the heuristic value in the following ways:
> TDH -- domain(detailed in Table 7 of the Appendix, same as below): The relevance of task domains to DSNN. If the current partial program does not align with the DSL's field, the heuristic value gets a negative gain.
> SSPH -- feasibility: The viability of synthesizing the complete program. If the current partial program cannot produce the complete program,  the heuristic value gets a negative gain.
> SGH -- other output features: Characteristics that rules for partial programs should meet. Consistency provides a positive gain for heuristic value, while inconsistency results in a negative gain.
> I can expand on these details in the Appendix.
>
> --------------------------------------------------------------------------
> Now, let me address your specific questions:
>
> 1 How are the neural networks trained?
>
> The training methodology employs continuous learning, as illustrated in the Introduction (about dynamic experiments) and Experiment (such as Figure 6,7 captions). If you are focusing on the training data structure,  it aligns with the neural network structure detailed in Figure 9 of the Appendix, which clearly outlines the input-output. Appendix O includes a sample of the training data. The entire project code, along with a complete training dataset, can be downloaded from the attachment. If you think it is necessary, I can list the training part in a separate section in the Appendix.
>
> I think your concern might be how is the training data structured for training those three heads of DSNN? Please refer to my answer to question 7.
>
> 2 How are the heuristics computed?
>
> The heuristic values are proved in the CoL heuristic vectors by programmers. The way NNFC affects heuristic values ​​has been answered previously.
>
> 3 What are the input and output for a given synthesis problem?
>
> The synthesis process occurs at the intermediate representation (IR) level. Given the IR's low readability, the complete program synthesis for the relational reasoning problem is presented as: relation = "brother". For symbolic problems in Appendices L and M, the synthesis outputs an equation like a*x^2 + b*x + c == 0, which can leverage the solution formula directly. Although the COOL syntax is not discussed due to the ICLR theme, the provided examples are intended to be comprehensible. For detailed syntax, please visit the anonymous link coologic.org.
>
> 4 Quadratic equation dataset
>
> Appendices L and M showcase the quadratic equation problems addressed. Specific datasets are not disclosed in the Appendix but are available for download in the attachment.

---

> ### Author Response · Authors · 2024-11-14
>
> 5 Overview of the algorithm that transforms input partial program to output program
>
> Algorithm D synthesizes partial programs into complete algorithms using u2 (refer to Figure 4) as the reward/cost value. While this algorithm is not central to the paper, it serves as a control variable, facilitating the paper's reproducibility. The core focus is on CoL and NNFC as the control structures for DSL program synthesis, rather than a specific algorithm.
>
> 6 What are tree operations and transformation pairs? How is the search process conducted?
>
> Tree operations involve changes in the syntax tree from applying DSL rules to partial programs, directly influencing synthesis efficiency. A transformation pair consists of a partial program and the rule applied to it. The synthesis path/trajectory maps the transformation pairs from the initial partial program to the final complete program, represented as [(P0 R0) (P1 R1) ... (complete P/ failure/ other)], where P signifies a program and R signifies a rule. The essence of DSL program synthesis is to explore this path efficiently, and neural network learning focuses on (P R). While these terms are widely recognized in DSL synthesis, I can clarify definitions in the appendix if needed, and your argument about "the evaluation is not very convincing" is not persuasive.
>
> 7 How is data collected?
>
> As noted in question 6, the data (P R) is automatically exported. Including a detailed discussion of the export process may overly engineer the paper and detract from its computational focus. Likewise, in the referenced papers on reinforcement learning DSL program synthesis, the data collection process was not elaborated in detail when the project code was provided.
> A data set for training a DSNN for DSL "A" is as follows: (P R) of other DSLs for training Task Detection Head (TDH),  (P R) of DSL "A" but in infeasible synthesis paths for training Search Space Prune Head (SSPH), (P R) of DSL "A" and in feasible synthesis pathes for training Search Guidance Head (SGH). I will provide an overview of the export process in the Appendix if necessary.
>
> -----------------------------------------------------------
>
> I believe our research holds significant value for DSL program synthesis. The theoretical foundation of our work is robust and has undergone rigorous experimental validation. I am willing to enhance the content in the appendix according to your expectations and provide additional references to specific implementations within the main text of the paper.
>
> Furthermore, I feel that your evaluation should focus on the technological advancements, contributions, and experimental rigor of our work rather than assigning a low score solely based on the complexity or interdisciplinary nature of the paper. I appreciate your time in reviewing the submission and wish you to share ANY confusion or concerns you may have. Thank you.

---

> > ### Comment · Reviewer_7Vwc · 2024-11-26
> >
> > Thank you for your response. I appreciate your attempts to clarify what's going on in the paper, but I can't justify increasing my score given the obfuscated presentation of the material, and the fact that given what I can understand from the paper, the results do not seem very impressive to me.

---

### Official Review · Reviewer_AVwJ · 2024-11-05

**Soundness:** 1
**Presentation:** 1
**Contribution:** 1
**Rating:** 1
**Confidence:** 5

**Summary:**

The submission discusses a technique for synthesis of logical rules from data.

**Strengths:**

n/a

**Weaknesses:**

The paper fails at presenting the contribution, and I am not able to provide a review of the actual contribution due to the presentation.

After reading the paper, I do not know
* what the form of the actual inputs to the system are, and how a user would interact with it
* how the components of the system are constructed (i.e., what is "DSNN" - is it a feed-forward network? An LLM? What shape does its input take? What is its actual output? Similarly, THD, SSPH, SGH are only explained at the highest possible level)
* what ideas can transfer to other contexts

The only way forward I see is to rewrite the entire submission. I would recommend to ask colleagues who are not closely familiar with this project to review draft submissions, and provide feedback on whether they understand what you are trying to present, before submitting the paper again.

**Questions:**

n/a

---

> ### Author Response · Authors · 2024-11-13
> **Clarifying Confusion and hoping you can review it again.**
>
> I admit that the paper is complex, but all content is well-reasoned and clearly presented. I understand that your confusion may stem from the lack of references to the appendix within the main text sections of the paper. I will ensure to provide these references for clarity and hope that you can spend more time reviewing this paper:
>
> Firstly, in the "Summary," you mentioned, "The submission discusses a technique for synthesis of logical rules from data." Upon review, the paper focuses on DSL program synthesis, where rules are employed to transform a partial program into a complete one. This focus is clearly articulated in the Abstract, Introduction, and Method sections, rather than centering on data synthesis rules as initially stated.
>
> In the "Weakness", you said that the paper was poorly presented and that you failed to capture the actual contribution. We understand that the technical complexity of the paper may have contributed to this impression. Here are some clarifications to address your confusion:
>
> 1. what the form of the actual inputs to the system are, and how a user would interact with it
>
> Answer: Firstly, Figure 1 on the first page of this paper illustrates the distinction between our DSL program synthesis system and traditional DSL program synthesis systems. It clearly marks the input and output in the figure. Specifically, the input to a DSL program synthesis system is a partial program (a program with non-terminal elements), while the output is a complete program (a program without non-terminal elements). This distinction is also detailed in Section 2.1 on page 4. Additionally, in the experiments section, the tasks utilized are explicitly defined, with the actual code provided in Appendices J through N.
>
> 2. how the components of the system are constructed (i.e., what is "DSNN" - is it a feed-forward network? An LLM? What shape does its input take? What is its actual output? Similarly, THD, SSPH, SGH are only explained at the highest possible level)
>
> Answer: Including abstract methods in the main body of the paper and specific implementations in the appendix is a common approach, especially when dealing with complex content. This paper employs a spatiotemporal graph attention network. Detailed information about the specific network structure, inputs and outputs, and the dimensions of each layer is provided in Appendix C (NEURAL NETWORKS IN DSNN) through accompanying figures and tables.  The program format of the direct processing of the neural network is shown in Appendix O (PARTIAL PROGRAM AS NEURAL NETWORK INPUT)
>
> 3 what ideas can transfer to other contexts?
>
> Answer: Given the confusion regarding the main research content of this paper, I am uncertain about what you mean by "idea" and "other context." The core research content of this paper centers on DSL program synthesis, which involves transforming partial programs with non-terminal symbols into complete programs consisting solely of terminal symbols through the application of rules. The key improvement lies in enhancing the efficiency of rule application.
> We introduce the CoL (Chain-of-Logic) approach to decompose the rule application stage and provide heuristic guidance. Additionally, we enhance the adaptability of CoL through Neural Network Feedback Control (NNFC). Our method's capability in handling complex DSL program synthesis tasks—including increasing problem difficulty and executing multi-DSL collaboration tasks—is confirmed through a series of experiments.
>
>
> Although the paper is complex, this is necessary for a thorough discussion. I encourage you to spend more time reviewing it and ask any questions you may have. I am confident in its innovation and contribution, and I believe I can help you recognize the significant advancements in DSL program synthesis technology presented in this paper.
>
>
>
> -------------------------
> For more about DSL program synthesis:
> 1 ICLR papers closely related to this paper:
> Neural Guided Constraint Logic Programming for Program Synthesis,
> ExeDec: Execution Decomposition for Compositional Generalization in Neural Program Synthesis
> Anonymous link to the project: coologic.org

---

> > ### Comment · Reviewer_AVwJ · 2024-11-14
> > **Review Discussion**
> >
> > > I admit that the paper is complex, but all content is well-reasoned and clearly presented.
> >
> > We all sometimes get unlucky with our reviewers, but 4 independent reviews that all highlight the poor presentation should give you pause, and encourage you to reflect on whether there may be an actual issue here. In your replies to me and the other reviewers, you refer to the appendices for core definitions, but remember that the main paper is meant to be understandable by the ICLR audience (that isn't necessarily familiar with Program Synthesis or Logic Programming) without reference to the appendices.
> >
> > Here are some examples of core information that is missing from the main text of the paper:
> > 1. What are the actual stages of the synthesis procedure that are claimed as a contribution of CoL, i.e., what can we express as a stage (beyond using `return`, `logicjump` and `abort`)?
> > 2. What do the heuristic values mean? Why is $(0, 7, 3)$ a good choice in Fig. 1, and why would $(0, 3, 7)$ be a bad one? Could I use $(-1, 42, 23, -1)$ as a heuristic value?
> > 3. How are different expansions of the partial program actually explored? From appendix D.1, we can assume that this is A* search, but that's never explicitly stated. It remains unclear how that interacts with the defined stages. Algorithm 1 also refers to a function input $u_2$ - we can guess that this is always $u_2$ from Eq (3), but that is defined in terms of $u_1$, which is maybe the $u_1$ from Fig. 4 - or maybe not, as it's not defined in the paper.
> > 4. Fig. 4 refers to a "Series Neural Network", a term that doesn't appear anywhere outside of the figure in the paper. What does it mean? Does it just indicate that it's a NN that's used before or after another one?
> >
> > > the input to a DSL program synthesis system is a partial program (a program with non-terminal elements), while the output is a complete program (a program without non-terminal elements).
> >
> > Who writes the DSL? The examples that you provide in the paper (Fig. 1/3/8, Appendices H/I) indicate that the DSL is extremely specialised to the problem domain, and so can conceivably be considered part of the input. This is supported by the paper text, stating things like "As illustrated in Figure 1, programmers can precisely organize rules into multiple stages and manage control flow using heuristics and keywords.", which seems to indicate that the programmer is expected to provide these things as input.
> >
> > I'll also point out that the paper has some issues around the term DSL, e.g. stating "A DSL, defined as a context-free grammar, converts partial programs with nonterminal symbols into complete programs by applying given rules" - that's not the normal definition of a DSL, but of a system that operates on top of a DSL.
> >
> > > Including abstract methods in the main body of the paper and specific implementations in the appendix is a common approach
> >
> > You are not providing an abstract method - you are providing names, and single-line summaries. Things such as "the program is transformed into graph representation on which X, Y, Z operate" would be crucial here.
> >
> > > spatiotemporal graph attention network
> >
> > What is the "spatiotemporal" component here? What temporal dimension exists?
> >
> > > I am uncertain about what you mean by "idea" and "other context."
> >
> > We write and read research papers to share ideas and insights.
> >
> > The context in which you present your work _in your paper_, namely synthesis of rules around family trees, is not a problem that anyone cares about in itself - it's a proxy problem, studied because problems that are interesting to actual users outside of a research context are much harder, both for exposition and actually solving them.
> >
> > So the question then is: what insights did you develop during the work going into this paper that could be meaningfully applied outside a research context, or on substantially more complex problems?

---

> > > ### Author Response · Authors · 2024-11-14
> > >
> > > ```
> > >  We all sometimes get unlucky with our reviewers, but 4 independent reviews that all highlight the poor presentation should give you pause, and encourage you to reflect on whether there may be an actual issue here.
> > > ```
> > > I noticed this, but I also noticed that the variance in the scores is so large. In fact, the third reviewer gave it a 6 at first, and he also understood the content of the paper accurately and asked in-depth questions.
> > >
> > > In contrast, your initial review suggested that you may not have fully engaged with the paper after a brief look-through. I understand that reviewing is a voluntary task, and this approach can be time-efficient for papers requiring only minor revisions. However, this particular paper is complex and demands a more thorough reading.
> > >
> > > Given the constraints of a 10-page limit for the main text, it is common in research to present high-level technical content and experimental findings concisely while relegating detailed information—such as algorithms, examples, and datasets—to the appendix. This practice is especially relevant for papers that cover extensive material.
> > >
> > > You indicated that the main body of the paper should be able to stand alone from the appendix. While I agree with this principle, it is important to consider that the main text may only provide a broad overview. Readers seeking specific details or clarification may find it necessary to consult the appendix. Therefore, it would be more constructive to examine the appendix for comprehensive information rather than concluding that "the author did not present this."
> > >
> > >
> > > ```
> > > What are the actual stages of the synthesis procedure that are claimed as a contribution of CoL, i.e., what can we express as a stage (beyond using return, logicjump and abort)?
> > > ```
> > > The CoL decomposes the synthesis process from a single stage into several smaller stages. It is equivalent to a complex task being decomposed into multiple steps, not proposing a new program synthesis process.
> > >
> > > ```
> > > What do the heuristic values mean?
> > > ```
> > > I use an example in Section 2.1 to demonstrate that heuristic values are the elements in a heuristic vector. You can use $(-1,42,23,-1)$ as a heuristic vector, and the heuristic values in stages 1,2,3,4 is -1,42,23,-1, respectively. Heuristic values ​​are used in heuristic algorithms, and their specific usage and effect depend on the algorithm. If the algorithm treats the heuristic value as a reward or weight, the rule with a large heuristic value will be more likely to be used; if it is a cost or penalty, the opposite is true. You can check the wiki for details. The paper uses the A* heuristic algorithm and treats it as a reward (see Appendix D.1).
> > >
> > > ```
> > > How are different expansions of the partial program actually explored?
> > > ```
> > > Yes, the expansions of a partial program are explored in the A* algorithm, and the lowest cost(negative reward) synthesis path is explored preferably. This paper focuses on CoL and NNFC structure. Therefore, the algorithm for the DSL solver is a control variable in this paper. The stages divided by CoL are a constraint for valid rule application, influencing the searching space. Different partial programs may stay in various stages with different searching spaces. Without considering keywords, the stages can only progress (Appendix B). u2 in Algorithm 1 is the u2 in Figure 4. Algorithm 1 signifies how the solver of CoL DSL works, and Equation(3) shows how Clipper in Figure 4 works. They are all control variables that aren't detailed in the main text.
> > >
> > > ```
> > >  "Series Neural Network": Does it just indicate that it's an NN that's used before or after another one?
> > > ```
> > > Yes, they simply refer to the series or parallel connection of the structure. For specific connection state, please see Table 6
> > >
> > > ```
> > > Who writes the DSL?
> > > ```
> > > The "expert" familiar with the domain-specific problems writes the DSL, and general users use the DSL. Only the users' programs are regarded as inputs. In other words, you can think DSL is a kind of library and the library with CoL + NNFC is better to use. Normally this is not a confusing point in DSL programming synthesis, but I can add a section in the appendix to sort out all the inputs and outputs in detail.
> > >
> > > ```
> > > I'll also point out that the paper has some issues around the term DSL, e.g. stating "A DSL, defined as a context-free grammar, converts partial programs with nonterminal symbols into complete programs by applying given rules" - that's not the normal definition of a DSL, but of a system that operates on top of a DSL.
> > > ```
> > > This is the complete version: A DSL, defined as a context-free grammar $G=(V, \Sigma, R, S)$, where V is the set of non-terminal symbols, $\Sigma$ is the set of terminal symbols, R is the set of rules, and S is the starting symbol (in this context is a partial program). DSL's derivation process) converts partial programs with nonterminal symbols into complete programs by applying given rules.

---

> ### Author Response · Authors · 2024-11-14
>
> ```
> Including abstract methods in the main body of the paper and specific implementations in the appendix is a common approach
> ```
> What this means is that the main text is written in an abstract, high-level, generalized manner, not specifically referring to algorithms or methods (sorry for your confusion)
>
> ```
> spatiotemporal graph attention network
> ```
> A description of a composite network consisting of graph attention layer and recurrent layer (the network described in Appendix C belongs to this category)
>
> ```
> what insights did you develop during the work going into this paper that could be meaningfully applied outside a research context, or on substantially more complex problems?
> ```
>
> As mentioned in the conclusion, We believe that our research will inspire advancements in broader areas of neural network reasoning. For instance, in our dynamic experiments, the neural network within the NNFC loop is trained from scratch, and its predictive ability improves through continuous training on synthesis paths during batch execution of synthesis tasks. In this process, the neural network's performance can be unreliable, but our program synthesis system demonstrates high reliability. In fact, program synthesis is a kind of neural network reasoning. As problem complexity increases, the number of steps rises, making it essential to minimize error probabilities at each stage. Therefore, I confirm promoting the technology presented in this paper will stimulate further research into neural network reasoning, particularly for complex tasks.
>
>
>
> Finally, let me briefly explain the relationship between acquiring neural network training data and the DSL program synthesis task. This explanation is crucial for reducing the complexity of reading the paper, and I will also organize it separately in the appendix.
>
> In DSL, a complete program is synthesized by applying rules to partial programs. Each partial program (P) and the corresponding rules (R) form a transformation pair (P, R). When a rule is applied, it modifies the syntax tree of the partial program through what we refer to as a tree operation. The complete synthesis process consists of a series of transformation pairs connected by tree operations, represented as [(P0, R0), (P1, R1), ... (complete P/failure/other)], which is known as a synthesis path or trajectory.
>
> The essence of the DSL synthesis process is a search problem focused on efficiently constructing this path. To learn from earlier experience, the neural network leverages these transformation pairs (P, R) on synthesis paths for supervised learning, with P used as the input and R as the label.
>
>
> If you have any more questions, please let me know.

---

> ### Comment · Reviewer_AVwJ · 2024-11-15
> **Review Discussion**
>
> > Therefore, it would be more constructive to examine the appendix for comprehensive information rather than concluding that "the author did not present this."
>
> Please refer to the ICLR call for papers on https://iclr.cc/Conferences/2025/CallForPapers:
> "Authors may use as many pages of appendices (after the bibliography) as they wish, but reviewers are not required to read the appendix."
>
> I am happy to look into the appendix for specific details, but I reject your notion that it's the reviewers job to scour the appendix to make the main paper text understandable.
>
> >> What are the actual stages of the synthesis procedure that are claimed as a contribution of CoL, i.e., what can we express as a stage (beyond using return, logicjump and abort)?
>
> > The CoL decomposes the synthesis process from a single stage into several smaller stages. It is equivalent to a complex task being decomposed into multiple steps, not proposing a new program synthesis process.
>
> I asked precise question ("what can we express as a stage (beyond using return, logicjump and abort)") that you did not address. Decomposition into objects of undescribed structure or power is not a meaningful answer; it's the same as saying that "the synthesis process is decomposed into several small frumbles. Each frumble is simpler than the original task".
>
> You claim this decomposition as a contribution of the paper, but you fail to explain what a stage is. We can guess that it's maybe a set of relations from Fig. 1, or maybe something that looks like an imperative program (as the "detailed rule"), but we do not know what expressions are allowed here - can we have a loop, for example?
>
> [As a side note, looking at Fig. 1 again, I finally realised that the first stage has a rule of the shape `f(x, y) -> g(y, x), h(x)`, and the subsequent rules have the shape `f(x, y) := g(y, x)` - is that `->` vs. `:=` difference a typo? Meaningful?]
>
> >> What do the heuristic values mean?
>
> > I use an example in Section 2.1 to demonstrate that heuristic values are the elements in a heuristic vector. You can use as a heuristic vector, and the heuristic values in stages 1,2,3,4 is -1,42,23,-1, respectively.
>
> So here you indicate that the individual scalars in a heuristic vector correspond to the defined stages, but in Fig. 1 we see that a rule is annotated with the heuristic value `(0, 7, 3)`. So what does that mean? Given that rules in the "CoL DSL" in Fig. 1 are named like the stages in Fig. 3, I assumed that the rules correspond to the stages, but this is in conflict with your statement here, and the "For example, in Figure 1, a rule with the heuristic vector (0,7,3) is applicable in stages 2 and 3 with heuristic
> values of 7 and 3, respectively" in the paper.
> This goes back to the fact that you never define what a stage actually is, or how this would be defined in the DSL.
>
> Finally, you have not actually explained what the heuristic values are used for. I can guess that it's maybe used to determine which rules should be preferred in the search procedure (and that higher values make a rule more likely to be explored), but this is never stated.
>
> I am fully convinced that the paper in its current form is not fit for publication at ICLR, and that it requires a major revision that is not in scope for the conference reviewing process. Something in the preparation of this paper went very wrong, and you need to reflect on how you can make sure that a revised version is understandable by your target audience.

---

> > ### Author Response · Authors · 2024-11-15
> >
> > ```
> > I asked precise question ("what can we express as a stage (beyond using return, logicjump and abort)") that you did not address. Decomposition into objects of undescribed structure or power is not a meaningful answer; it's the same as saying that "the synthesis process is decomposed into several small frumbles. Each frumble is simpler than the original task".
> > ```
> >
> > This question is like how to define the objects generated by the "divide and conquer strategy". I cannot give a precise answer to this abstract question. But to clarify your doubts, I think I can answer it like this:
> >
> > The original program synthesis process is divided into multiple sequential stages. In each stage, the intermediate form of the program generated from the earlier stage is regarded as the input of the next stage, which is handled by the sub-DSL of the corresponding stage. The rules of each sub-DSL are selected from the set of rules of the original DSL, and the selection is based on the fact that the specific bit in the heuristic vector of the rule is not zero.
> >
> > ```
> > [As a side note, looking at Fig. 1 again, I finally realised that the first stage has a rule of the shape f(x, y) -> g(y, x), h(x), and the subsequent rules have the shape f(x, y) := g(y, x) - is that -> vs. := difference a typo? Meaningful?]
> > ```
> >
> > No. -> is left to right; := is right to left:
> >  f(x, y) -> g(y, x), h(x) means: replace f(x, y) with g(y, x), h(x) , while  f(x, y) := g(y, x), h(x) means: replace  g(y, x), h(x) with f(x,y)
> > [comma symbols "and"]
> >
> >
> > ```
> > So here you indicate that the individual scalars in a heuristic vector correspond to the defined stages, but in Fig. 1 we see that a rule is annotated with the heuristic value (0, 7, 3). So what does that mean? Given that rules in the "CoL DSL" in Fig. 1 are named like the stages in Fig. 3, I assumed that the rules correspond to the stages, but this is in conflict with your statement here, and the "For example, in Figure 1, a rule with the heuristic vector (0,7,3) is applicable in stages 2 and 3 with heuristic values of 7 and 3, respectively" in the paper. This goes back to the fact that you never define what a stage actually is, or how this would be defined in the DSL.
> > ```
> > I think the earlier answer to how to choose rules has made it clear.
> >
> > If you have any other confusion, please let me know. After all, all my answers are altered ways of original expressions in the paper. You said reviewers have the right not to read the Appendix. I respect that, but I don't recommend not doing this when there is confusion.

---

### Note · Authors · 2025-01-13

I have read and agree with the venue's withdrawal policy on behalf of myself and my co-authors.